# Personalized Federated Learning with Feature Alignment and Classifier Collaboration

**Jian Xu, Xinyi Tong, Shao-Lun Huang**[*]
Tsinghua Shenzhen International Graduate School, Tsinghua University

## Abstract

Data heterogeneity is one of the most challenging issues in federated learning, which motivates a variety of approaches to learn personalized models for participating clients. One such approach in deep neural networks based tasks is employing a shared feature representation and learning a customized classifier head for each client. However, previous works do not utilize the global knowledge during local representation learning and also neglect the fine-grained collaboration between local classifier heads, which limit the model generalization ability. In this work, we conduct explicit local-global feature alignment by leveraging global semantic knowledge for learning a better representation. Moreover, we quantify the benefit of classifier combination for each client as a function of the combining weights and derive an optimization problem for estimating optimal weights. Finally, extensive evaluation results on benchmark datasets with various heterogeneous data scenarios demonstrate the effectiveness of our proposed method.

## 1 Introduction

Modern learning tasks are usually enabled by deep neural networks (DNNs), which require huge quantities of training data to achieve satisfied model performance (Lecun et al., 2015; Krizhevsky et al., 2012; Hinton et al., 2012). However, collecting data is too costly due to the increasingly large volume of data or even prohibited due to privacy protection. Hence, developing communication-efficient and privacy-preserving learning algorithms is of significant importance for fully taking advantage of the data in clients, e.g., data silos and mobile devices (Yang et al., 2019; Li et al., 2020a). To this end, federated learning (FL) emerged as an innovative technique for collaborative model training over decentralized clients without gathering the raw data (McMahan et al., 2017). A typical FL setup employs a central server to maintain a global model and allows partial client participation with infrequent model aggregation, e.g., the popular FedAvg, which has shown good performance when local data across clients are independent and identically distributed (IID). However, in the context of FL, data distributions across clients are usually not identical (non-IID or heterogeneity) since different devices generate or collect data separately and may have specific preferences, including feature distribution drift, label distribution skew and concept shift, which make it hard to learn a single global model that applies to all clients (Zhao et al., 2018; Zhu et al., 2021a; Li et al., 2022).

To remedy this, personalized federated learning (PFL) has been developed, where the goal is to learn a customized model for each client that has better performance on local data while still benefiting from collaborative training (Kulkarni et al., 2020; Tan et al., 2021a; Kairouz et al., 2021). Such settings can be motivated by cross-silo FL, where autonomous clients (e.g., hospitals and corporations) may wish to satisfy client-specific target tasks. A practical FL framework should be aware of the data heterogeneity and flexibly accommodate local objectives during joint training. On the other hand, the DNNs based models are usually comprised of a feature extractor for extracting low-dimensional feature embeddings from data and a classifier[1] for making a classification decision. The success of deep learning in centralized systems and multi-task learning demonstrates that the feature extractor plays the role of common structure while the classifier tends to be highly task-correlated (Bengio et al., 2013; Collins et al., 2021; Caruana, 1993). Moreover, clients in practical FL problems often deal with similar learning tasks and clustered structure among clients is assumed in many prior works (Ghosh et al., 2020; Sattler et al., 2021). Hence, learning a better global feature representation and exploiting correlations between local tasks are of significant importance for improving personalized models.

---

[*]Corresponding author: shaolun.huang@sz.tsinghua.edu.cn

[1]We denote *classifier head* as the final output layer for decision-making, and abbreviate it to *classifier*.

In this work, we mainly consider the label distribution shift scenario, where the number of classes is the same across clients while the number of data samples in each class has obvious drift, i.e., heterogeneous label distributions for the local tasks. We study federated learning from a multi-task learning perspective by leveraging both shared representation and inter-client classifier collaboration. Specifically, we make use of the global feature centroid of each class to regularize the local training, which can be regarded as explicit feature alignment and is able to reduce the representation diversity across locally trained feature extractors, thus facilitating the global aggregation. We also conduct flexible classifier collaboration through client-specific linear combination, which encourages similar clients to collaborate more and avoids negative transfer from unrelated clients. To estimate the proper combining weights, we utilize local feature statistics and data distribution information to achieve the best bias-variance trade-off by solving a quadratic programming problem that minimizes the expected testing loss for each client. Moreover, with a slight modification, our framework could still work well under the concept shift scenario, where the same label might have varied meanings across clients.

**Our contributions.** We focus on deep learning-based classification tasks and propose a novel FL framework equipped by personalized aggregation of classifiers (**FedPAC**) and feature alignment to improve the overall performance of client-specific tasks. The proposed framework is evaluated on benchmark datasets with various levels of data heterogeneity to verify the effectiveness in achieving higher model performance. Our evaluation results demonstrate the proposed method can improve the average model accuracy by 2∼5%. To summarize, this paper makes the following key contributions:

- We quantify the testing loss for each client under the classifier combination by characterizing the discrepancy between the learned model and the target data distribution, which illuminates a new bias-variance trade-off.

- A novel personalized federated learning framework with feature representation alignment and optimal classifier combination is proposed for achieving fast convergence and high model performance.

- Through extensive evaluation on real datasets with different levels of data heterogeneity, we demonstrate the high adaptability and robustness of FedPAC.

**Benefits of FedPAC.** The benefits of our method over current personalized FL approaches include:

(i) More local updates for representation learning. By leveraging feature alignment to control the drift of local representation learning, each client can make many local updates with less local-global parameter diversity at each communication round, which is beneficial in learning better representation in a communication-efficient manner.

(ii) Gains by classifier heads collaboration. We employ a theoretically guaranteed optimal weighted averaging for combing heads from similar clients, which is capable of improving generalization ability for data-scarce clients while preventing negative knowledge transfer from unrelated clients.

## 2 RELATED WORK

### 2.1 FEDERATED LEARNING WITH NON-IID DATA

Many efforts have been devoted to improving the global model learning of FL with non-IID data. A variety of works focus on optimizing local learning algorithms by leveraging well-designed objective regularization (Li et al., 2020b; Acar et al., 2021a; Li et al., 2021b) and local bias correction (Karimireddy et al., 2020b;a; Murata & Suzuki, 2021; Dandi et al., 2021). For example, FedProx (Li et al., 2020b) adds a proximal term to the local training objective to keep updated parameter close to the original downloaded model, SCAFFOLD (Karimireddy et al., 2020b) introduces control variates to correct the drift in local updates, and MOON (Li et al., 2021b) adopts the contrastive loss to improve the representation learning. Class-balanced data re-sampling and loss re-weighting methods can improve the training performance when clients have imbalanced local data (Hsu et al., 2020; Wang et al., 2021; Chen & Chao, 2021). Besides, data sharing mechanisms and data augmentation methods are also investigated to mitigate the non-IID data challenges (Zhao et al., 2018; Yoon et al., 2021). From the model aggregation perspective, selecting clients with more contribution to global model performance can also speed up the convergence and mitigate the influence of non-IID data (Wang et al., 2020; Tang et al., 2021; Wu & Wang, 2021; Fraboni et al., 2021). With the availability of public data, it is possible to employ knowledge distillation techniques to obtain a global model despite of the data heterogeneity (Lin et al., 2020; Zhu et al., 2021b). The prototype-based methods are also utilized in some FL works, such as (Michieli & Ozay, 2021) proposes a prototype-based weight

attention mechanism during global aggregation and (Mu et al., 2021; Tan et al., 2021b; Ye et al., 2022; Zhou et al., 2022) utilize the prototypes to enhance the local model training. Different from above methods, this paper aims at learning a customized model for each client.

## 2.2 MODEL PERSONALIZATION IN FL

In the literature, popular personalized FL methods include additive model mixture that performs linear combination of local and global modes, such as L2CD (Hanzely & Richtárik, 2020) and APFL (Deng et al., 2020); multi-task learning with model dissimilarity penalization, including FedMTL (Smith et al., 2017), pFedMe (Dinh et al., 2020) and Ditto (Li et al., 2021c); meta-learning based local adaption (Jiang et al., 2019; Fallah et al., 2020; Acar et al., 2021b); parameter decoupling of feature extractor and classifier, such as FedPer, LG-FedAvg and FedRep (Arivazhagan et al., 2019; Liang et al., 2020; Collins et al., 2021). A special type of personalized FL approach is clustered FL to group together similar clients and learn multiple intra-group global models (Duan et al., 2021; Ghosh et al., 2020; Mansour et al., 2020; Sattler et al., 2021). Client-specific model aggregations are also investigated for fine-grained federation, such as FedFomo and FedAMP (Zhang et al., 2021b; Huang et al., 2021; Beaussart et al., 2021), which have similar spirit to our approach. Nevertheless, existing client-specific FL methods are usually developed by evaluating model similarity or validation accuracy in a heuristic way, and these techniques need to strike a good balance between communication/computation overhead and the effectiveness of personalization. FedGP based on Gaussian processes (Achituve et al., 2021) and selective knowledge transfer based solutions (Zhang et al., 2021a; Cho et al., 2021) are also developed, however those methods inevitably rely on public shared data set or inducing points set. Besides, pFedHN enabled by a server-side hypernetwork (Shamsian et al., 2021) and FedEM that learns a mixture of multiple global models (Marfoq et al., 2021) are also investigated for generating customized model for each client. However, pFedHN requires each client to communicate multiple times for learning a representative embedding and FedEM significantly increases both communication and computation/storage overhead. Recently, Fed-RoD (Chen & Chao, 2022) proposes to use the balanced softmax for learning generic model and vanilla softmax for personalized heads. FedBABU (Oh et al., 2022) proposes to keep the global classifier unchanged during the feature representation learning and perform local adoption by fine-tuning. kNN-Per (Marfoq et al., 2022) applies the ensemble of global model and local kNN classifiers for better personalized performance. Our work shares the most similar learning procedure with FedRep (Collins et al., 2021), but differs in the sense that we employ global knowledge for guiding local representation learning and also perform theoretically-guaranteed classifier heads combination for each client.

## 3 OVERVIEW OF PROPOSED FRAMEWORK

In this section, we provide an overview of our approach FedPAC for training personalized models by exploiting a better feature extractor and user-specific classifier collaboration. Theoretical justification and algorithm design will be presented in the next two sections.

### 3.1 PROBLEM SETUP

We consider a setup with $m$ clients and a central server, where all clients communicate to the server to collaboratively train personalized models without sharing raw private data. In personalized FL, each client $i$ is equipped with its own data distribution $P_{XY}^{(i)}$ on $\mathcal{X} \times \mathcal{Y}$, where $\mathcal{X}$ is the input space and $\mathcal{Y}$ is the label space with $K$ categories in total. We assume that $P_{XY}^{(i)}$ and $P_{XY}^{(j)}$ are different for any pair of client $i$ and $j$, which is usually the case in FL. Let $\ell : \mathcal{X} \times \mathcal{Y} \to \mathbb{R}_+$ denote the loss function given local model $\boldsymbol{w}_i$ and data point sampled from $P_{XY}^{(i)}$, e.g., cross-entropy loss, then the underlying optimization goal of PFL can be formalized as follows:

$$\min_{\mathbf{W}} \left\{ F(\mathbf{W}) := \frac{1}{m} \sum_{i=1}^{m} \mathbb{E}_{(x,y) \sim P_{XY}^{(i)}} [\ell(\boldsymbol{w}_i; x, y)] \right\} \tag{1}$$

where $\mathbf{W} = (\boldsymbol{w}_1, \boldsymbol{w}_2, ..., \boldsymbol{w}_m)$ denotes the collection of all local models. However, the true underlying distribution is inaccessible and the goal is usually achieved by empirical risk minimization (ERM). Assume each client has access to $n_i$ IID data points sampled from $P_{XY}^{(i)}$, denoted by $\mathcal{D}_i = \{(x_l^{(i)}, y_l^{(i)})\}_{l=1}^{n_i}$, whose corresponding empirical distribution is $\hat{P}_{XY}^{(i)}$[2], and we assume the

---

[2]The empirical distribution is defined as $\hat{P}_{XY}^{(i)}(x, y) := \frac{1}{n_i} \sum_{l=1}^{n_i} \mathbb{1} \left\{ x = x_l^{(i)}, y = y_l^{(i)} \right\}$

empirical marginal distribution $\hat{P}_Y^{(i)}$ is identical to the true $P_Y^{(i)}$. Then the training objective is

$$\boldsymbol{w}^* = \arg\min_{\mathbf{w}} \frac{1}{m} \sum_{i=1}^m \left[ \mathcal{L}_i(\boldsymbol{w}_i) + \mathcal{R}_i(\boldsymbol{w}_i; \boldsymbol{\Omega}) \right] \tag{2}$$

where $\mathcal{L}_i(\boldsymbol{w}_i) = \frac{1}{n_i} \sum_{l=1}^{n_i} \ell(\boldsymbol{w}_i; x_l^{(i)}, y_l^{(i)})$ is the local average loss over personal training data, e.g., empirical risk; $\boldsymbol{\Omega}$ is some kind of global information introduced to relate clients, and the $\mathcal{R}_i(\cdot)$ is a predefined regularization term for preventing $\boldsymbol{w}_i$ from over-fitting.

## 3.2 SHARING FEATURE REPRESENTATION

Without loss of generality, we decouple the deep neural network into the representation layers and the final decision layer, where the former is also called feature extractor, and the latter refers to the classifier head in classification tasks. The feature embedding function $\boldsymbol{f} : \mathcal{X} \to \mathbb{R}^d$ is a learnable network parameterized by $\boldsymbol{\theta}_f$ and $d$ is the dimension of feature embedding. Given a data point $x$, a prediction for extracted feature vector $z = \boldsymbol{f}(x)$ can be generated by a linear function $\boldsymbol{g}(z)$ parameterized by $\phi_g$. In the rest of paper, we omit the subscripts of $\boldsymbol{\theta}_f$ and $\phi_g$ for simplicity, i.e., $\boldsymbol{w} = \{\boldsymbol{\theta}, \phi\}$. Since each client only has insufficient data, the locally learned feature representation is prone to over-fitting and thus cannot generalize well. A reasonable and feasible idea is to take advantage of data available at other clients by sharing the same feature representation layers. However, multiple local updates on private data will result in local over-fitting and high parameter diversity across clients, which make the aggregated model deviate from the best representation. To tackle this, we propose a new regularization term for local feature representation learning.

**Feature Alignment by Regularization.** The clients need to update the local model by considering both supervised learning loss and the generalization error. To this end, we leverage the global feature centroids and introduce a new regularization term to the local training objective to enable the local representation learning benefit from global data. The local regularization term is given by

$$\mathcal{R}_i(\boldsymbol{\theta}_i; \boldsymbol{c}) = \frac{\lambda}{n_i} \sum_{l=1}^{n_i} \frac{1}{d} \left\| \boldsymbol{f}_{\boldsymbol{\theta}_i}(x_l) - \boldsymbol{c}_{y_l} \right\|_2^2 \tag{3}$$

where $\boldsymbol{f}_{\boldsymbol{\theta}_i}(x_j)$ is the local feature embedding of given data point $x_j$, and $\boldsymbol{c}_{y_j}$ is the corresponding global feature centroid of class $y_j$, $\lambda$ is hyper-parameter to balance supervised loss and regularization loss. Such a regularization term is of significant benefit to each client by leveraging global semantic feature information. Intuitively, it enables each client to learn task-invariant representations through explicit feature distribution alignment. As a result, the diversity of local feature extractors $\{\theta_i\}_{i=1}^m$ could also be regularized while minimizing local classification error. We notice that the global feature centroid is similar to the concept of prototype that is widely used in few-shot learning, contrastive learning and domain adaption (Snell et al., 2017; Li et al., 2021a; Pan et al., 2019). Our theoretical insight in Section 4.2 will further reveal that this regularization term could be inspired by explicitly reducing generalization error and thus helps to improve the test accuracy.

## 3.3 CLASSIFIER COLLABORATION

In addition to improving feature extractor by sharing representation layers, we argue that merging classifiers from other clients with similar data distribution can also offer performance gain. Unlike previous work in (Arivazhagan et al., 2019; Collins et al., 2021) that only maintains the locally trained classifier, we also conduct a client-specific weighted average of classifiers to obtain a better local classifier. Intuitively, the locally learned classifier may have a high variance when the local data is insufficient, therefore those clients that share similar data distribution can actually collaboratively train the personalized classifiers by inter-client knowledge transfer. The challenge is how to evaluate the similarity and transferability across clients. To this end, we conduct a linear combination of those received classifiers for each client $i$ to reduce the local testing loss:

$$\hat{\phi}_i^{(t+1)} = \sum_{j=1}^m \alpha_{ij} \phi_j^{(t+1)}, \quad s.t. \sum_{j=1}^m \alpha_{ij} = 1 \tag{4}$$

with each coefficient $\alpha_{ij} \geq 0$ determined by minimizing local expected testing loss, which can be formulated as the following optimization problem:

$$\boldsymbol{\alpha}_i^* = \arg\min_{\boldsymbol{\alpha}_i} \mathbb{E}_{(x,y)\sim P_{XY}^{(i)}}[\ell(\boldsymbol{\theta}, \sum_{j=1}^m \alpha_{ij}\boldsymbol{\phi}_j; x, y)] \tag{5}$$

For a better collaboration, we need to update the coefficients $\boldsymbol{\alpha}_i$ adaptively during the training process.

**Remark 1.** *Consider a simple case where data are homogeneous across clients. In such a setting, learning a single global model is optimal for generalization. However, some existing methods, such as FedPer and FedRep, do not take into account the combination of classifier heads and thus lead to limited performance in less heterogeneous cases. In contrast, our method conducts an adaptive classifier combination and is effective in both homogeneous and heterogeneous scenarios.*

## 4 THEORETICAL ANALYSIS AND INSIGHTS

In this section, we introduce the theoretical analyses of our proposed approach for improving personalized federated learning from both the feature extractor and classifier perspectives. Specifically, we consider the framework introduced in Section 3.2, where the local model of each client contains the shared feature extractor $\boldsymbol{f}$ and a personalized linear classifier $\boldsymbol{g}_i$ parameterized by $\boldsymbol{w}_i = (\boldsymbol{\theta}, \boldsymbol{\phi}_i)$, respectively. For the convenience of theoretical analysis and evaluation, we choose the following linear discriminative model[3]:

$$Q_{Y|X}(y|x; \boldsymbol{w}) = \frac{1}{K}\left(1 + \boldsymbol{f}(x)^{\mathrm{T}}\boldsymbol{g}(y)\right), \tag{6}$$

which has the similar formation as the linear model used in (Xu & Huang, 2020; Tong et al., 2021).

Note that usually model parameters are updated in an end-to-end manner according to the general training objective (2). However, when the classifier collaboration is discussed, we will concentrate on the scenario where the feature extractor is fixed. Then, we simply use $Q_{Y|X}(\boldsymbol{g})$ to denote the model and learn the individual classifiers $\{\boldsymbol{g}_i\}_{i=1}^m$ of different clients and linearly combine them for each client $i$ with the weights $\boldsymbol{\alpha}_i$. Further, we introduce the $\chi_i^2$-distance in (7), which is different from the conventional chi-square divergence and is defined as follows. For any distributions $P$ and $Q$ supported on $\mathcal{X} \times \mathcal{Y}$, the $\chi_i^2$-distance between $P$ and $Q$ is given by

$$\chi_i^2(P, Q) := \sum_{x\in\mathcal{X}, y\in\mathcal{Y}} \frac{(P(x,y) - Q(x,y))^2}{P_X^{(i)}(x)}, \tag{7}$$

where $P_X^{(i)}$ is the marginal distribution over $\mathcal{X}$. Here we select this distance measure as the loss function mainly in consideration of the interpretability and deriving the analytical solution.

For this discriminative model, ideally, the local model could be derived by minimizing the $\chi_i^2$-distance between $P_{XY}^{(i)}$ and $P_X^{(i)}Q_{Y|X}^{(i)}$, which refer to the true local data distribution and the product of marginal distribution and the learned model, respectively. However, each client will only access limited training samples and the locally learned classifier might not work well enough, which directly motivates the inter-client collaboration. Since the $\boldsymbol{f}$ and $\boldsymbol{g}$ are highly correlated during the model learning, jointly analyzing them is difficult. Instead, we fix either $\boldsymbol{f}$ or $\boldsymbol{g}$ by turns and apply the analysis-inspired method to optimize the learning of the unfixed one in an iterative manner.

### 4.1 OPTIMAL CLASSIFIER COMBINATION

Under the fixed shared feature extractor $\boldsymbol{f}$, we can empirically learn a classifier $\hat{\boldsymbol{g}}_i$ based on the $\boldsymbol{f}$ and training samples at client $i$ in the following way:

$$\hat{\boldsymbol{g}}_i := \arg\min_{\boldsymbol{g}} \chi_i^2(\hat{P}_{XY}^{(i)}, P_X^{(i)}Q_{Y|X}^{(i)}(\boldsymbol{g})), \quad \forall i \in [m] \tag{8}$$

Then, we can use a linear combination by $\hat{\boldsymbol{g}}_i^* = \sum_{j=1}^m \alpha_{ij}\hat{\boldsymbol{g}}_j$ to substitute for the original $\hat{\boldsymbol{g}}_i$ as illustrated in Section 3.3. Similar to the training loss in (8), analytically we use the $\chi^2(\cdot, \cdot)$ over true data distribution as the testing loss to evaluate the performance of resulted personalized classifier. Let $\boldsymbol{\alpha}_i := (\alpha_{i1}, \cdots, \alpha_{im})^{\mathrm{T}}$ and the testing loss for client $i$ is given by

$$R_i(\boldsymbol{\alpha}_i) := \mathbb{E}\left[\chi_i^2\left(P_{XY}^{(i)}, P_X^{(i)}Q_{Y|X}^{(i)}(\hat{\boldsymbol{g}}_i^*)\right)\right], \tag{9}$$

---

[3]This linear model has the same prediction criteria as the softmax classifier and could be interpreted as the approximation of softmax model with a large temperature on the logits.

where the expectation is taken over the sampling process, i.e., empirical distributions $\{\hat{P}_{XY}^{(i)}\}_{i=1}^{m}$ that follow the multinomial distributions (Csiszár, 1998).

Before performing theoretical analyses, we assume that underlying marginals over $\mathcal{X}$ are identical across clients and the extracted feature embedding is zero-centered as follows.

**Assumption 1.** For all $i \in [m]$, we have $P_X^{(i)}(x) = P_X(x)$.

**Assumption 2.** For the given feature extractor $\boldsymbol{f}$, we have $\mathbb{E}_{P_X}[\boldsymbol{f}(X)] = \boldsymbol{0}$.

Assumption 1-2 are mainly technical to derive our analytical solution and are not strictly required for our method to work in practice. In the general cases, determining the combining weights of classifiers trained by other clients requires to learn the generative models. However, under the identical marginal assumption we can focus on the discriminative models. When the whole input space is accessible, we could apply zero-centering to satisfy the zero-mean feature assumption. Now, we can characterize the testing loss for each client $i$ under the linear combination of classifiers by the following theorem.

**Theorem 1.** *The testing loss for client $i$ as defined in (9) is a quadratic function w.r.t. $\boldsymbol{\alpha}_i$ given by*

$$R_i(\boldsymbol{\alpha}_i) = \chi_i^2\left(P_X^{(i)}Q_{Y|X}^{(i)}(\boldsymbol{g}_i), \sum_{j=1}^{m}\alpha_{ij}P_X^{(i)}Q_{Y|X}^{(i)}(\boldsymbol{g}_j)\right) + \sum_{j=1}^{m}\frac{\alpha_{ij}^2}{n_j}V_j + \chi_i^2\left(P_{XY}^{(i)}, P_X^{(i)}Q_{Y|X}^{(i)}(\boldsymbol{g}_i)\right) \tag{10}$$

*where for all $j \in [m]$, $\boldsymbol{g}_j := \mathbb{E}[\hat{\boldsymbol{g}}_j]$ and $V_j$ is a variance term that related to the local feature statistics. Details and proofs are provided in Appendix A.2.*

We argue that the established testing loss is related to the bias-variance trade-off in the personalized classifiers, where the first term in (10) refers to the bias term of the model utilizing other clients and the second term refers to the variances from sampling, which are shown in $\sum_{j=1}^{m}\alpha_{ij}^2 V_j/n_j$. We utilize the classifiers in other clients to reduce the variance, and meanwhile it increases the bias to the true distribution $P_{XY}^{(i)}$. To achieve the optimal bias-variance trade-off, we define the best combination coefficients as

$$\boldsymbol{\alpha}_i^* := \arg\min_{\boldsymbol{\alpha}_i} R_i(\boldsymbol{\alpha}_i), \quad s.t. \quad \sum_{j=1}^{m}\alpha_{ij} = 1 \quad and \quad \alpha_{ij} \geq 0, \forall j \tag{11}$$

Note that the terms appearing in the testing loss are related to the true distributions, which can be estimated by the corresponding empirical feature statistics, with details provided in Appendix A.3. When the optimal $\boldsymbol{\alpha}_i^*$ is estimated, we can realize our proposed personalized classifier as (4).

## 4.2 REDUCING TESTING LOSS BY FEATURE ALIGNMENT

Given a fixed classifier head $\boldsymbol{g}_i$ and dataset $\mathcal{D}_i$ at client $i$, the testing loss can be rewritten as follows:

$$R_i = \chi_i^2\left(\hat{P}_{XY}^{(i)}, P_X^{(i)}Q_{Y|X}^{(i)}(\boldsymbol{g}_i)\right) + \sum_{x,y} 2\left(P_{XY}^{(i)}(x,y) - \hat{P}_{XY}^{(i)}(x,y)\right)\boldsymbol{f}(x)^{\mathrm{T}}\boldsymbol{g}_i(y)/K + C_i \tag{12}$$

where $C_i$ is a constant and it is easy to see that the first term is the empirical loss and the remainder can be viewed as generalization gap. Due to the discrepancy between true underlying distribution $P_{XY}^{(i)}$ and empirical distribution $\hat{P}_{XY}^{(i)}$, minimizing training loss could result in large generalization gap when training data is insufficient. On the other hand, the second term in (12) can be rewritten by

$$\Delta R_i = 2\,\mathbb{E}_{P_Y^{(i)}}\left[\mathbb{E}_{P_{X|Y}^{(i)}}[\boldsymbol{f}(x)] - \mathbb{E}_{\hat{P}_{X|Y}^{(i)}}[\boldsymbol{f}(x)]\right]^{\mathrm{T}}\boldsymbol{g}_i(Y)/K \tag{13}$$

Benefiting from FL, the true conditional distribution $P_{X|Y}^{(i)}$ could be better estimated by taking advantage of data available at other clients, and therefore the above term can be further estimated by local empirical data $\mathcal{D}_i$ and global feature centroids $\boldsymbol{c}$ as follows,

$$\Delta\hat{R}_i = \frac{2}{n_i}\sum_{l=1}^{n_i}[\boldsymbol{c}_{y_l} - \boldsymbol{f}(x_l)]^{\mathrm{T}}\boldsymbol{g}_i(y_l)/K \tag{14}$$

where $\boldsymbol{c}_{y_l}$ is the feature centroid of class $y_l$ estimated by collecting local feature statistics. Intuitively, this error term (14) is directly induced by the inconsistency of local-global feature representations,

which means the learned feature distribution is not compact. However, there exists a trade-off between training loss and generalization gap, and simultaneously minimizing them is infeasible. Therefore, similar to (Tan et al., 2021b), we design a regularization term as Eq. (3) during the feature learning phase to explicitly align the local-global feature representations.

## 5 Algorithm Design

In this section, we design a alternating optimization approach to iteratively learn the local classifier and global feature extractor. Locally updated model parameters along with feature statistics will be transmitted to the central server for aggregation (more details provided in Appendix A.5).

### 5.1 Local Training Procedure

For local model training at round $t$, we first replace the local representation layers $\boldsymbol{\theta}_i^{(t)}$ by the received global aggregate $\tilde{\boldsymbol{\theta}}^{(t)}$ and update private classifier analogously. Then, we perform stochastic gradient decent steps to train the two parts of model parameters as follows:

• **Step 1: Fix $\boldsymbol{\theta}_i$, Update $\boldsymbol{\phi}_i$** Train $\boldsymbol{\phi}_i$ on private data by gradient descent for one epoch:

$$\boldsymbol{\phi}_i^{(t)} \leftarrow \boldsymbol{\phi}_i^{(t)} - \eta_g \nabla_{\boldsymbol{\phi}} \ell(\boldsymbol{\theta}_i^{(t)}, \boldsymbol{\phi}_i^{(t)}; \xi_i) \tag{15}$$

where $\xi_i$ denotes the mini-batch of data, $\eta_g$ is the learning rate for updating classifier.

• **Step 2: Fix New $\boldsymbol{\phi}_i$, Update $\boldsymbol{\theta}_i$** After getting new local classifier, we turn to train the local feature extractor based on both private data and global centroids for multiple epochs:

$$\boldsymbol{\theta}_i^{(t)} \leftarrow \boldsymbol{\theta}_i^{(t)} - \eta_f \nabla_{\boldsymbol{\theta}} \left[ \ell(\boldsymbol{\theta}_i^{(t)}, \boldsymbol{\phi}_i^{(t+1)}; \xi_i) + \mathcal{R}_i(\boldsymbol{\theta}_i^{(t)}; \boldsymbol{c}^{(t)}) \right] \tag{16}$$

where $\eta_f$ is the learning rate for updating representation layers, $\boldsymbol{c}^{(t)} \in \mathbb{R}^{K \times d}$ is the collection of global feature centriod vector for each class, and $K = |\mathcal{Y}|$ is the total number of classes.

**Feature Statistics Extraction:** Before local feature extractor updating, each client should extract the local feature statistics $\boldsymbol{\mu}_i^{(t)}$ and $V_i^{(t)}$ through a single pass on the local dataset, which will be utilized to estimate the optimal classifier combination weights for each client. Moreover, after updating the local feature extractor, we compute the local feature centroid for each class as follows,

$$\hat{\boldsymbol{c}}_{i,k}^{(t+1)} = \frac{\sum_{l=1}^{n_i} \mathbf{1}(y_l^{(i)} = k) \boldsymbol{f}_{\boldsymbol{\theta}_i^{(t+1)}}(x_l^{(i)})}{\sum_{l=1}^{n_i} \mathbf{1}(y_l^{(i)} = k)}, \forall k \in [K]. \tag{17}$$

### 5.2 Global Aggregation

**Global Feature Representation.** Like the common algorithms, the server performs weighted averaging of local representation layers with each coefficient determined by the local data size.

$$\tilde{\boldsymbol{\theta}}^{(t+1)} = \sum_{i=i}^{m} \beta_i \boldsymbol{\theta}_i^{(t)}, \quad \beta_i = \frac{n_i}{\sum_{i=1}^{m} n_i}. \tag{18}$$

**Classifier Combination.** The server uses received feature statistics to updates the combination weights vector $\boldsymbol{\alpha}_i$ by solving (11) and conducts classifier combination for each client $i$.

**Update Global Feature Centroids.** After receiving the local feature centroids, the following centroid aggregating operation is conducted to generate an estimated global centroid $\boldsymbol{c}_k$ for each class $k$.

$$\boldsymbol{c}_k^{(t+1)} = \frac{1}{\sum_{i=1}^{m} n_{i,k}} \sum_{i=1}^{m} n_{i,k} \hat{\boldsymbol{c}}_{i,k}^{(t+1)}, \quad \forall k \in [K]. \tag{19}$$

## 6 Experiments

### 6.1 Experimental Setup

**Datasets and Models.** We consider image classification tasks and evaluate our method on four popular datasets: EMNIST with 62 categories of handwritten characters, Fashion-MNIST with 10 categories of clothes, CIFAR-10 and CINIC-10 with 10 categories of color images. We construct two

different CNN models for EMNIST/Fashion-MNIST and CIFAR-10/CINIC-10, respectively. Details of datasets and model architectures are provided in Appendix B.

**Data Partitioning.** Similar to (Karimireddy et al., 2020b; Zhang et al., 2021b; Huang et al., 2021), we make all clients have the same data size, in which $s\%$ of data (20% by default) are uniformly sampled from all classes, and the remaining $(100 - s)\%$ from a set of dominant classes for each client. We explicitly divide clients into multiple groups while clients in each group share the same dominant classes, and we also intentionally keep the size of local training data small to pose the need for FL. The testing data on each client has the same distribution as the training data.

**Compared Methods.** We compare the following baselines: Local-only, where each client trains its model locally; FedAvg that learns a single global model and its locally fine-tuned version (FedAvg-FT); multi-task learning methods, including APFL, pFedMe and Ditto; parameter decoupling methods, including LG-FedAvg, FedPer, FedRep and FedBABU; Fed-RoD and kNN-Per that learn an extra local classifier based on the global feature extractor and use model ensemble for prediction; FedFomo that conducts inter-client linear combination and pFedHN enabled by a server-side hypernetwork.

**Training Settings.** We employ the mini-batch SGD as a local optimizer for all approaches, and the number of local training epochs is set to $E = 5$ unless explicitly specified. The number of global communication rounds is set to 200 for all datasets, where all FL approaches have little or no accuracy gain with more communications. We report the average test accuracy across clients.

Table 1: The comparison of final test accuracy (%) on different datasets. We apply full participation for FL system with 20 clients, and apply client sampling with rate 0.3 for FL system with 100 clients.

| Method | EMNIST | | Fashion-MNIST | | CIFAR-10 | | CINIC-10 | |
|---|---|---|---|---|---|---|---|---|
| | 20 clients | 100 clients | 20 clients | 100 clients | 20 clients | 100 clients | 20 clients | 100 clients |
| Local-only | 73.85 | 74.11 | 85.68 | 86.37 | 65.43 | 64.68 | 63.19 | 63.33 |
| FedAvg | 71.85 | 75.97 | 85.28 | 86.89 | 70.05 | 73.82 | 58.38 | 62.52 |
| FedAvg-FT | 84.56 | 87.37 | 90.47 | 91.53 | 76.56 | 79.34 | 69.38 | 74.44 |
| FedPer | 75.91 | 77.06 | 87.43 | 87.88 | 68.37 | 72.26 | 63.47 | 66.26 |
| LG-FedAvg | 74.76 | 75.81 | 85.66 | 86.28 | 65.19 | 65.38 | 63.75 | 63.93 |
| FedRep | 75.31 | 75.60 | 88.25 | 88.54 | 71.36 | 72.58 | 66.67 | 66.96 |
| FedBABU | 83.52 | 85.59 | 89.68 | 91.38 | 76.46 | 78.39 | 70.58 | 74.67 |
| Fed-RoD | 81.94 | 84.57 | 90.11 | 91.61 | 77.16 | 81.97 | 71.82 | 75.20 |
| kNN-Per | 80.21 | 83.15 | 90.02 | 90.93 | 76.73 | 80.47 | 69.98 | 74.23 |
| APFL | 83.22 | 84.95 | 89.45 | 90.88 | 74.39 | 77.32 | 68.78 | 73.45 |
| pFedMe | 83.28 | 84.89 | 89.76 | 89.85 | 68.81 | 69.85 | 63.37 | 65.31 |
| Ditto | 84.21 | 86.61 | 90.43 | 91.08 | 77.02 | 80.09 | 70.14 | 74.25 |
| FedFomo | 83.95 | 85.63 | 88.95 | 90.19 | 73.33 | 76.21 | 66.68 | 70.22 |
| pFedHN | 80.35 | 78.16 | 88.36 | 87.93 | 76.95 | 79.39 | 71.59 | 70.85 |
| **FedPAC** | **86.46** | **89.88** | **91.83** | **92.72** | **81.13** | **83.36** | **74.96** | **77.36** |

## 6.2 NUMERICAL RESULTS

**Performance Comparison.** We conduct experiments on two setups, where the number of clients is 20 and 100, respectively. For the latter, we apply random client selection with sampling rate $C$=0.3 along with full participation in the last round. The training data size in each client is set to 600 for all datasets except EMNIST, where the size is 1000. The main results are presented in Table 1, it is obvious that our proposed method performs well on both small-scale and large-scale FL systems. For all datasets, FedPAC dominates the other methods on average test accuracy, which demonstrates the effectiveness and benefit of global feature alignment and inter-client classifier collaboration. The classifier combination weights are shown in Appendix C.1. We also notice that FedAvg with local fine-tuning is already a strong baseline for PFL, resulting in competitive performance as the start-of-the-art methods.

**Ablation Studies.** We have two key design components in FedPAC, i.e., feature alignment (FA) and classifier combination (CC). Here we conduct ablation studies to verify the individual efficacy of those two components. We make use of FA and CC separately on four datasets and report the average accuracy on 20 clients. As demonstrated in Table 2, both of them can help improve the average test accuracy and the combination of them is able to achieving the most satisfactory model performance, which means a better global feature extractor and more suitable personalized classifiers could be built under our proposed method.

Table 2: Ablation study. **FA** denotes Feature Alignment and **CC** denotes Classifier Collaboration; **None** means neither FA nor CC is used, while **Both** means FA and CC are applied simultaneously.

| Dataset | Design Choices in FedPAC | | | |
|---|---|---|---|---|
| | None | w/ FA | w/ CC | Both |
| EMNIST (%) | 75.63 | 86.45 | 82.78 | **86.46** |
| Fashion-MNIST (%) | 87.93 | 89.74 | 89.92 | **91.83** |
| CIFAR-10 (%) | 71.68 | 79.43 | 78.19 | **81.13** |
| CINIC-10 (%) | 66.20 | 72.89 | 70.84 | **74.96** |

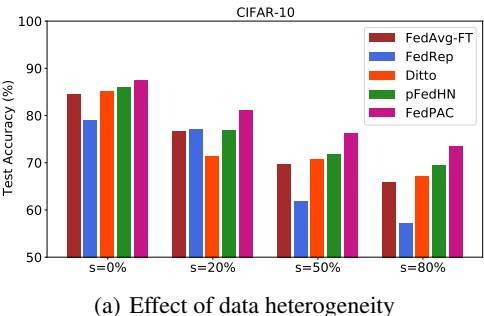
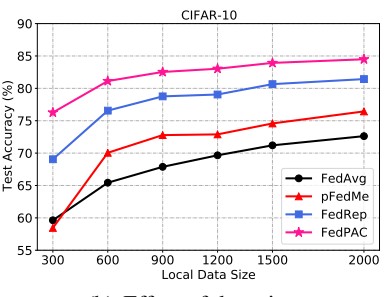

(a) Effect of data heterogeneity

(b) Effect of data size

Figure 1: Performance comparison on CIFAR-10 dataset with varying data heterogeneity and local data size.

**Effects of Data Heterogeneity and Data Size.** We vary the values of $s$ to simulate different levels of data heterogeneity, while $s$=0% indicates a highly heterogeneous case (pathological non-IID) and $s$=80% means data across clients are more homogeneous. We evaluate the CIFAR-10 dataset and the results of different methods are reported in Figure 1 (a). It can be found that our method consistently outperforms other baselines, which demonstrates its adaptability and robustness in a variety of heterogeneous data scenarios. We also test our FedPAC and FedAvg with varying local data sizes, recording the resulted model accuracy as in Figure 1 (b). The results indicate that clients with different data sizes can consistently benefit from participating in FL and our method achieves higher performance gain.

**Robustness to Concept Shift.** Though the concept shift is not our focus, we show that our method is also applicable and robust in such scenarios. To simulate the concept shift scenario, each group of clients will perform a specific permutation of labels. The results in Table 3 show that the label permutation hurts the performance of FedAvg-like algorithms that learn a global classifier as different learning tasks are conflicting. On the other hand, the general improvement over Local-only demonstrates that leveraging the global feature extractor can still benefit from data among clients with concept shift.

Table 3: Test accuracy (%) under concept shift. We apply full client participation for FL system with 20 clients.

| Dataset | Local-only | FedAvg-FT | FedRep | Ditto | pFedMe | FedBABU | Fed-RoD | **FedPAC** |
|---|---|---|---|---|---|---|---|---|
| Fashion-MNIST | 85.68 | 87.38 | 87.12 | 87.78 | 86.84 | 87.28 | 87.60 | **91.09** |
| CIFAR-10 | 65.43 | 70.55 | 68.12 | 70.41 | 62.71 | 70.43 | 69.44 | **78.45** |

# 7 CONCLUSION AND FUTURE WORK

In this paper, we introduce the global feature alignment for enhancing representation learning and a novel classifier combination algorithm for building personalized classifiers in FL, providing both theoretical and empirical justification for their utilities in heterogeneous settings. Future work includes analyzing the optimal model personalization in more complex settings, e.g., decentralized systems or clients with dynamic data distributions and investigating the optimal aggregation of local feature extractors.

## ACKNOWLEDGEMENTS

The research of Shao-Lun Huang is supported in part by the Shenzhen Science and Technology Program under Grant KQTD20170810150821146, National Key R&D Program of China under Grant 2021YFA0715202.

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

APPENDIX

# A    PROOFS OF THEORETICAL RESULTS

## A.1    RELATION BETWEEN $\chi^2$-DISTANCE AND K-L DIVERGENCE

$$L_i'(\boldsymbol{g}) = \sum_{x \in \mathcal{X}} P_X^{(i)}(x) \sum_{y \in \mathcal{Y}} P_{Y|X}^{(i)}(y|x) \log \frac{P_{Y|X}^{(i)}(y|x)}{Q_{Y|X}^{(i)}(y|x)}. \tag{20}$$

With $\log(1+x) \sim x - \frac{x^2}{2}$, $P_{Y|X}^{(i)}(y|x) \approx P_Y(y)$,

$$L_i'(\boldsymbol{g}) \approx \frac{1}{2} \sum_{x \in \mathcal{X}} P_X^{(i)}(x) \sum_{y \in \mathcal{Y}} \frac{(P_{Y|X}^{(i)}(y|x) - Q_{Y|X}^{(i)}(y|x))^2}{P_{Y|X}^{(i)}(y|x)} \tag{21}$$

$$\approx \frac{1}{2} \sum_{x \in \mathcal{X}} P_X^{(i)}(x) \sum_{y \in \mathcal{Y}} \frac{(P_{Y|X}^{(i)}(y|x) - Q_{Y|X}^{(i)}(y|x))^2}{P_Y(y)} \tag{22}$$

$$= \frac{1}{2} \sum_{x \in \mathcal{X}} \sum_{y \in \mathcal{Y}} \frac{P_X^{(i)2}(x)(P_{Y|X}^{(i)}(y|x) - Q_{Y|X}^{(i)}(y|x))^2}{P_X^{(i)}(x)P_Y(y)} \tag{23}$$

$$= \frac{1}{2} \sum_{x \in \mathcal{X}} \sum_{y \in \mathcal{Y}} \frac{(P_X^{(i)}(x)P_{Y|X}^{(i)}(y|x) - P_X^{(i)}(x)Q_{Y|X}^{(i)}(y|x))^2}{P_X^{(i)}(x)P_Y(y)} \tag{24}$$

$$\propto \chi_i^2(P_{XY}^{(i)}, P_X^{(i)}Q_{Y|X}^{(i)}(\boldsymbol{g})) \tag{25}$$

## A.2    PROOF OF THEOREM 1

For better readability, we denote $P_Y(y) = \frac{1}{K}$ the uniform distribution over the label space. We first express the $\hat{\boldsymbol{g}}_i$. Since we have

$$\chi_i^2 \left( \hat{P}_{XY}^{(i)}, P_X^{(i)}Q_{Y|X}^{(i)}(\boldsymbol{g}) \right)$$

$$= \sum_{x \in \mathcal{X}, y \in \mathcal{Y}} \frac{1}{P_X^{(i)}(x)} \cdot \left( \hat{P}_{XY}^{(i)}(x,y) - P_X^{(i)}(x)P_Y(y) - P_X^{(i)}(x)P_Y(y)\boldsymbol{f}^{\mathrm{T}}(x)\boldsymbol{g}(y) \right)^2, \tag{26}$$

then, for all $y' \in \mathcal{Y}$,

$$\frac{\partial \chi_i^2 \left( \hat{P}_{XY}^{(i)}, P_X^{(i)}Q_{Y|X}^{(i)}(\boldsymbol{g}) \right)}{\partial \boldsymbol{g}(y')}$$

$$= -2 \sum_{x \in \mathcal{X}} \hat{P}_{XY}^{(i)}(x,y')P_Y(y')\boldsymbol{f}(x) + 2 \sum_{x \in \mathcal{X}} P_X^{(i)}(x)[P_Y(y')]^2 \left( 1 + \boldsymbol{f}^{\mathrm{T}}(x)\boldsymbol{g}(y') \right) \boldsymbol{f}(x)$$

$$= -2 \sum_{x \in \mathcal{X}} \hat{P}_{XY}^{(i)}(x,y')P_Y(y')\boldsymbol{f}(x) + 2[P_Y(y')]^2 \boldsymbol{\Lambda}_{\boldsymbol{f}}^{(i)}\boldsymbol{g}(y'), \tag{27}$$

where to obtain the last equality, we have used the assumption that $\mathbb{E}_{P_X^{(i)}}[\boldsymbol{f}(X)] = \boldsymbol{0}$ (in implementation, we can apply zero-centering to satisfy this assumption) and the notation that $\boldsymbol{\Lambda}_{\boldsymbol{f}}^{(i)} \triangleq \mathbb{E}_{P_X^{(i)}}[\boldsymbol{f}(X)\boldsymbol{f}^{\mathrm{T}}(X)]$.

Set the gradient (27) to zero, and we obtain

$$\hat{\boldsymbol{g}}_i(y) = \frac{1}{P_Y(y)}[\boldsymbol{\Lambda}_{\boldsymbol{f}}^{(i)}]^{-1} \left( \sum_{x \in \mathcal{X}} \hat{P}_{XY}^{(i)}(x,y)\boldsymbol{f}(x) \right). \tag{28}$$

We can precisely express the testing error (9) as

$$
R_i(\boldsymbol{\alpha}_i) = \mathbb{E}\left[\chi_i^2\left(P_{XY}^{(i)}, P_X^{(i)} Q_{Y|X}^{(i)}(\hat{\boldsymbol{g}}_i^*)\right)\right]
$$

$$
= \chi_i^2\left(P_{XY}^{(i)}, \sum_{j=1}^m \alpha_{ij} P_X^{(i)} Q_{Y|X}^{(i)}(\boldsymbol{g}_j)\right) + \sum_{j=1}^m \alpha_{ij}^2 \underbrace{\mathbb{E}\left[\chi_i^2\left(P_X^{(i)} Q_{Y|X}^{(i)}(\hat{\boldsymbol{g}}_j), P_X^{(i)} Q_{Y|X}^{(i)}(\boldsymbol{g}_j)\right)\right]}_{\frac{1}{n_j} V_j},
$$

(29)

where $\boldsymbol{g}_j = \mathbb{E}[\hat{\boldsymbol{g}}_j]$ is the expectation over the data sampling process.

Next, for the first term, we have

$$
\chi_i^2\left(P_{XY}^{(i)}, \sum_{j=1}^m \alpha_{ij} P_X^{(i)} Q_{Y|X}^{(i)}(\boldsymbol{g}_j)\right) = \chi_i^2\left(P_{XY}^{(i)}, P_X^{(i)} Q_{Y|X}^{(i)}(\boldsymbol{g}_i)\right)
$$

$$
+ \chi_i^2\left(P_X^{(i)} Q_{Y|X}^{(i)}(\boldsymbol{g}_i), \sum_{j=1}^m \alpha_{ij} P_X^{(i)} Q_{Y|X}^{(i)}(\boldsymbol{g}_j)\right), \quad (30)
$$

which comes from the fact that for any $\boldsymbol{g}$, we have

$$
\sum_{x\in\mathcal{X}, y\in\mathcal{Y}} \frac{\left(P_{XY}^{(i)}(x,y) - [P_X^{(i)} Q_{Y|X}^{(i)}(\boldsymbol{g}_i)](x,y)\right)[P_X^{(i)} Q_{Y|X}^{(i)}(\boldsymbol{g})](x,y)}{P_X^{(i)}(x)}
$$

$$
= \sum_{x\in\mathcal{X}, y\in\mathcal{Y}} \left(P_{XY}^{(i)}(x,y) - [P_X^{(i)} Q_{Y|X}^{(i)}(\boldsymbol{g}_i)](x,y)\right) P_Y(y)\boldsymbol{f}^{\mathrm{T}}(x)\boldsymbol{g}(y)
$$

$$
= \sum_{x\in\mathcal{X}, y\in\mathcal{Y}} P_{XY}^{(i)}(x,y) P_Y(y)\boldsymbol{f}^{\mathrm{T}}(x)\boldsymbol{g}(y)
$$

$$
- \sum_{x\in\mathcal{X}, y\in\mathcal{Y}} P_X^{(i)}(x) P_Y(y)\boldsymbol{f}^{\mathrm{T}}(x) \frac{1}{P_Y(y)}[\boldsymbol{\Lambda}_{\boldsymbol{f}}^{(i)}]^{-1}\left(\sum_{x\in\mathcal{X}} P_{XY}^{(i)}(x,y)\boldsymbol{f}(x)\right) P_Y(y)\boldsymbol{f}^{\mathrm{T}}(x)\boldsymbol{g}(y)
$$

$$
= \frac{1}{K}\left(\mathbb{E}_{P_{XY}^{(i)}}[\boldsymbol{f}^{\mathrm{T}}(X)\boldsymbol{g}(Y)] - \mathbb{E}_{P_{XY}^{(i)}}[\boldsymbol{f}^{\mathrm{T}}(X)\boldsymbol{g}(Y)]\right)
$$

$$
= 0.
$$

(31)

Then, for the bias term in the RHS of (30), we have

$$
\chi_i^2\left(P_X^{(i)} Q_{Y|X}^{(i)}(\boldsymbol{g}_i), \sum_{j=1}^m \alpha_{ij} P_X^{(i)} Q_{Y|X}^{(i)}(\boldsymbol{g}_j)\right)
$$

$$
= \sum_{j=1}^m \sum_{j'=1}^m \alpha_{ij} \alpha_{ij'} \sum_{x,y} P_X^{(i)}(x)[P_Y(y)]^2(\boldsymbol{f}^{\mathrm{T}}(x)\boldsymbol{g}_i(y) - \boldsymbol{f}^{\mathrm{T}}(x)\boldsymbol{g}_j(y))(\boldsymbol{f}^{\mathrm{T}}(x)\boldsymbol{g}_i(y) - \boldsymbol{f}^{\mathrm{T}}(x)\boldsymbol{g}_{j'}(y))
$$

$$
= \sum_{j=1}^m \sum_{j'=1}^m \alpha_{ij} \alpha_{ij'} D_{jj'},
$$

(32)

where by Assumption 1, we further have

$$
D_{jj'} = \sum_{x,y} P_X^{(i)}(x)[P_Y(y)]^2(\boldsymbol{f}^{\mathrm{T}}(x)\boldsymbol{g}_i(y) - \boldsymbol{f}^{\mathrm{T}}(x)\boldsymbol{g}_j(y))(\boldsymbol{f}^{\mathrm{T}}(x)\boldsymbol{g}_i(y) - \boldsymbol{f}^{\mathrm{T}}(x)\boldsymbol{g}_{j'}(y))
$$

$$
= \mathrm{tr}\left([\boldsymbol{\Lambda}_{\boldsymbol{f}}^{(i)}]^{-1} \sum_{y\in\mathcal{Y}} (h(i,y) - h(j,y))(h(i,y) - h(j',y))^{\mathrm{T}}\right),
$$

(33)

and

$$
h(l,y) \triangleq P_Y^{(l)}(y) \mathbb{E}_{P_{X|Y=y}^{(l)}}[\boldsymbol{f}(X)], \, l = 1, \cdots, m.
$$

(34)

Finally, for the variance term in (29), we have

$$
\begin{aligned}
V_j &= n_j\, \mathbb{E}\left[\chi_i^2\left(P_X^{(i)}Q_{Y|X}^{(i)}(\boldsymbol{g}_j),\, P_X^{(i)}Q_{Y|X}^{(i)}(\hat{\boldsymbol{g}}_j)\right)\right] \\
&= n_j\, \mathbb{E}\left[\sum_{x\in\mathcal{X},y\in\mathcal{Y}} P_X^{(i)}(x)[P_Y(y)]^2(\boldsymbol{f}^{\mathrm{T}}(x)\,(\boldsymbol{g}_j-\hat{\boldsymbol{g}}_j))^2\right] \\
&= n_j\, \mathbb{E}\left[\sum_{x\in\mathcal{X},y\in\mathcal{Y}} P_X^{(i)}(x)\left(\boldsymbol{f}^{\mathrm{T}}(x)[\boldsymbol{\Lambda}_{\boldsymbol{f}}^{(i)}]^{-1}\left(\sum_{x\in\mathcal{X}}\left(P_{XY}^{(j)}(x,y)-\hat{P}_{XY}^{(j)}(x,y)\right)\boldsymbol{f}(x)\right)\right)^2\right] \\
&= n_j \sum_{y=1}^{|\mathcal{Y}|}\mathrm{tr}\left([\boldsymbol{\Lambda}_{\boldsymbol{f}}^{(i)}]^{-1}\,\mathrm{Var}\left(\sum_{x\in\mathcal{X}}\left(P_{XY}^{(j)}(x,y)-\hat{P}_{XY}^{(j)}(x,y)\right)\boldsymbol{f}(x)\right)\right) \\
&= \sum_{y=1}^{|\mathcal{Y}|} P_Y^{(j)}(y)\,\mathrm{tr}\left([\boldsymbol{\Lambda}_{\boldsymbol{f}}^{(i)}]^{-1}\,\mathbb{E}_{P_{X|Y=y}^{(j)}}[\boldsymbol{f}(X)\boldsymbol{f}^{\mathrm{T}}(X)]\right) - [P_Y^{(j)}(y)]^2\left\|[\boldsymbol{\Lambda}_{\boldsymbol{f}}^{(i)}]^{-\frac{1}{2}}\,\mathbb{E}_{P_{X|Y=y}^{(j)}}[\boldsymbol{f}(X)]\right\|^2.
\end{aligned}
\tag{35}
$$

## A.3 Expressions of the quantities contained in (11)

In computing (11), after obtaining $\boldsymbol{f}$, we can estimate the quantities by the empirical expectations for each $i \in [m]$, including

$$
P_Y^{(i)}(y) \leftarrow \hat{P}_Y^{(i)}(y);
\tag{36}
$$

$$
\boldsymbol{\Lambda}_{\boldsymbol{f}}^{(i)} \leftarrow \hat{\boldsymbol{\Lambda}}_{\boldsymbol{f}}^{(i)} \triangleq \mathbb{E}_{\hat{P}_X^{(i)}}[\boldsymbol{f}(X)\boldsymbol{f}^{\mathrm{T}}(X)];
\tag{37}
$$

$$
\mathbb{E}_{P_{X|Y=y}^{(i)}}[\boldsymbol{f}(X)\boldsymbol{f}^{\mathrm{T}}(X)] \leftarrow \boldsymbol{\Lambda}_{i,y} \triangleq \mathbb{E}_{\hat{P}_{X|Y=y}^{(i)}}[\boldsymbol{f}(X)\boldsymbol{f}^{\mathrm{T}}(X)];
\tag{38}
$$

$$
\mathbb{E}_{P_{X|Y=y}^{(i)}}[\boldsymbol{f}(X)] \leftarrow \boldsymbol{\mu}_{i,y} \triangleq \mathbb{E}_{\hat{P}_{X|Y=y}^{(i)}}[\boldsymbol{f}(X)].
\tag{39}
$$

Now, the quantities contained in (11) could be estimated accordingly. Then, $\boldsymbol{\alpha}^*$ can be computed by solving a non-negative quadratic programming problem (e.g., using cvxpy toolbox).

## A.4 Expressions of the generalization gap in (14)

For fixed classifier head $\boldsymbol{g}_i$ and dataset $\mathcal{D}_i$ at client $i$, we have

$$
\begin{aligned}
R_i &= \chi_i^2\left(P_{XY}^{(i)},\, P_X^{(i)}Q_{Y|X}^{(i)}(\boldsymbol{g}_i)\right) \\
&= \chi_i^2\left(P_{XY}^{(i)}-\hat{P}_{XY}^{(i)}+\hat{P}_{XY}^{(i)},\, P_X^{(i)}Q_{Y|X}^{(i)}(\boldsymbol{g}_i)\right) \\
&= \sum_{x\in\mathcal{X},y\in\mathcal{Y}} \frac{\left(P_{XY}^{(i)}(x,y)-\hat{P}_{XY}^{(i)}(x,y)+\hat{P}_{XY}^{(i)}(x,y)-P_X^{(i)}(x)(1+\boldsymbol{f}(x)^{\mathrm{T}}\boldsymbol{g}_i(y))/K\right)^2}{P_X^{(i)}(x)} \\
&= \chi_i^2\left(\hat{P}_{XY}^{(i)},\, P_X^{(i)}Q_{Y|X}^{(i)}(\boldsymbol{g}_i)\right) + \underbrace{\sum_{x,y} 2\left(P_{XY}^{(i)}(x,y)-\hat{P}_{XY}^{(i)}(x,y)\right)\boldsymbol{f}(x)^{\mathrm{T}}\boldsymbol{g}_i(y)/K}_{\Delta R_i} + \underbrace{\chi_i^2\left(P_{XY}^{(i)},\hat{P}_{XY}^{(i)}\right)}_{C_i},
\end{aligned}
\tag{40}
$$

where the first term is the empirical loss and the last term is a constant $C_i$ that measures the discrepancy between the empirical data distribution and the true underlying data distribution at client

$i$. For the second term $\Delta R_i$, by assuming the empirical marginal distribution $\hat{P}_Y^{(i)}$ is identical to the true marginal distribution $P_Y^{(i)}$, we can further rewrite it by

$$\Delta R_i = 2 \sum_y P_Y^{(i)}(y) \left[ \sum_x \left( P_{X|Y}^{(i)}(x|y) - \hat{P}_{X|Y}^{(i)}(x|y) \right) \boldsymbol{f}(x) \right]^{\mathrm{T}} \boldsymbol{g}_i(y)/K$$

$$= 2 \, \mathbb{E}_{P_Y^{(i)}} \left[ \mathbb{E}_{P_{X|Y}^{(i)}}[\boldsymbol{f}(x)] - \mathbb{E}_{\hat{P}_{X|Y}^{(i)}}[\boldsymbol{f}(x)] \right]^{\mathrm{T}} \boldsymbol{g}_i(Y)/K, \tag{41}$$

where the empirical conditional distribution can be estimated by local dataset $\mathcal{D}_i$, while the true conditional distribution is unknown. However, profiting from the data available at other clients in FL, the true conditional distribution could be better estimated by calculating the global feature centroid of each class. Denote $\boldsymbol{c}_{y_l}$ the feature centroid of class $y_l$, we can further get the following estimate

$$\Delta \hat{R}_i = \frac{2}{n_i} \sum_{l=1}^{n_i} [\boldsymbol{c}_{y_l} - \boldsymbol{f}(x_l)]^{\mathrm{T}} \boldsymbol{g}_i(y_l)/K. \tag{42}$$

## A.5 PRACTICAL CONSIDERATION FOR ALGORITHM DESIGN

As shown in A.2 and A.3, to estimate the combination weights we need the covariance matrix of feature embeddings, which has the size of $d^2$ and could be large as the feature dimension $d$ increases. For the algorithm design, we make some slight modifications that do not affect the final performance. In consideration of the request of communication efficiency and privacy protection, we directly use the identity matrix to replace the feature covariance matrix of each client

$$\boldsymbol{\Lambda}_{\boldsymbol{f}}^{(i)} \leftarrow \boldsymbol{I}_d, \tag{43}$$

This operation is a compromise and can be inspired by the following fact. The cardinality $\mathcal{X}$ is a extremely large number and the feature $\boldsymbol{f}(x)$ is a low-dimensional vector. When the random noise (variance of all possible samples) added in the sample space (not Gaussian) is projected onto the low-dimensional space, the corresponding variance would be isotropic, which can be reflected as an identity matrix with a constant scaling and the scaling can be WLOG omitted in our calculation as it does not affect the results.

With this approximation, we can simplify the quantities (33) and (35) into

$$V_j \leftarrow \sum_{y=1}^{|\mathcal{Y}|} P_Y^{(j)}(y) \, \mathrm{tr} \left( \mathbb{E}_{P_{X|Y=y}^{(j)}} [\boldsymbol{f}(X)\boldsymbol{f}^{\mathrm{T}}(X)] \right) - [P_Y^{(j)}(y)]^2 \left\| \mathbb{E}_{P_{X|Y=y}^{(j)}}[\boldsymbol{f}(X)] \right\|^2, \tag{44}$$

and

$$D_{jj'} \leftarrow \mathrm{tr} \left( \sum_{y \in \mathcal{Y}} (h(i,y) - h(j,y))(h(i,y) - h(j',y))^{\mathrm{T}} \right). \tag{45}$$

By now, we can easily calculate the values of $V_j$ locally and get the values of $D_{jj'}$ by collecting $\boldsymbol{\mu}_l = \{\hat{P}_Y^{(l)}(y)\boldsymbol{\mu}_{l,y}\}_{y=1}^K$ from each client $l \in [m]$. Then, we can estimate the combination weights for each client by solving a quadratic programming problem (46) efficiently.

$$R_i(\boldsymbol{\alpha}_i) = \sum_{j=1}^m \frac{\alpha_{ij}^2}{n_j} V_j + \sum_{j=1}^m \sum_{j'=1}^m \alpha_{ij}\alpha_{ij'} D_{jj'}$$

$$= \sum_{j=1}^m \frac{\alpha_{ij}^2}{n_j} \sum_{y=1}^{|\mathcal{Y}|} \left[ \hat{P}_Y^{(j)}(y) \, \mathrm{tr}(\boldsymbol{\Lambda}_{j,y}) - \left\| \hat{P}_Y^{(j)}(y)\boldsymbol{\mu}_{j,y} \right\|^2 \right]$$

$$+ \sum_{j=1}^m \sum_{j'=1}^m \alpha_{ij}\alpha_{ij'} \, \mathrm{tr} \left( \sum_{y \in \mathcal{Y}} \left[ \hat{P}_Y^{(i)}(y)\boldsymbol{\mu}_{i,y} - \hat{P}_Y^{(j)}(y)\boldsymbol{\mu}_{j,y} \right] \left[ \hat{P}_Y^{(i)}(y)\boldsymbol{\mu}_{i,y} - \hat{P}_Y^{(j')}(y)\boldsymbol{\mu}_{j',y} \right]^{\mathrm{T}} \right) \tag{46}$$

---

**Algorithm 1** FedPAC

1: **Input:** Learning rate $\{\eta_f, \eta_g\}$, number of communication rounds $T$, number of clients $m$, number of local epochs $E$, hyper-parameter $\lambda$
2: **Server Executes:**
3: Initialize: $w^{(0)}$ and $c^{(0)}$
4: **for** $t = 0, 1, ..., T-1$ **do**
5:     Select client set $C_t$
6:     Broadcast $\{w^{(t)}, c^{(t)}\}$ to selected clients for local update
7:     Collect models and statistics from clients
8:     Get global feature extractor $\theta^{(t+1)}$ as (18) and global centriods $c^{(t+1)}$ as (19)
9:     Get personalized classifier $\tilde{\phi}_i^{(t+1)}$ by solving (11)
10:     Send $\{\theta^{(t+1)}, \tilde{\phi}_i^{(t+1)}\}$ back to client $i \in C_t$
11: **end for**
12: **ClientUpdate**$(i, w^{(t)}, c^{(t)})$:
13:     Update local feature extractor $\theta_i^{(t)} \leftarrow \tilde{\theta}^{(t)}$
14:     Extract feature statistics $\mu_i^{(t)}$ and $V_i^{(t)}$
15:     Train $\phi_i^{(t+1)}$ for 1 epoch and $\theta_i^{(t+1)}$ for $E$ epoch by turns
16:     Extract local feature centroids $\hat{c}_i^{(t+1)}$ as (17)
17:     Return $\{\theta_i^{(t+1)}, \phi_i^{(t+1)}, \hat{c}_i^{(t+1)}, \mu_i^{(t)}, V_i^{(t)}\}$ to server

---

It is also worth noting that the feature statistics used for estimating combination weights are calculated before local feature extractor training, while the class-wise feature centroids used for estimating global feature centroids are extracted after updating local feature extractors. By such a compromise, we can obtain a good estimate of updated global feature centroids without extra server-client communications. The main training procedure is summarized in Algorithm 1.

## A.6 Communication and Privacy Concerns

We acknowledge that FedPAC requires the transmission of local feature statistics between clients and server. However, this does not raise high privacy and communication issues since local feature statistics represent only an averaged statistic over all local data of low-dimensional feature representations. On the other hand, without privacy protection circumstances, it is hard to strictly guarantee the privacy. Even the vanilla FedAvg that only shares and aggregates local models would need more privacy-preserving techniques to enhance the system reliability in practice, e.g., homomorphic encryption (HE). With homomorphic encryption, the clients receive private keys to homomorphically encrypt their models before sending them to the server. The server doesn't own a key and only sees the encrypted models. But the server can aggregate these encrypted weights and then send the updated model back to the client. The clients can decrypt the model parameters because they have the keys and can then continue with the next round of local training and feature statistics extraction.

Now come back to the framework of FedPAC, where not only local models but also category-level feature statistics are transmitted, the aggregation of models can still be protected by applying HE techniques while the category-level feature statistics could be directly used to compute the personalized combination coefficients. We acknowledge that to further quantify the potential leakage from prototypes would involve in designing specific inversion attacks, which are out of the scope of our work. But we consider that the additional privacy risk may not be an issue when only category-level feature statistics are provided to server while the model parameters are encrypted. Without accessing the feature extractor layers, the server is hard to infer meaningful semantic data information from only the prototypes. And the privacy-preserving techniques are already equipped in many open-source FL libraries and widely applied in real-world applications.

Finally, we would like to point out that the trade-offs between utility, security and efficiency still exist in the context of personalized FL. Our framework mainly focuses on improving the model utility, for which additional communication and computation are required. We consider that the performance improvement is the main incentive for a client to participate the federated learning. In such cases, the clients may hope to exposure more information for better collaboration.

# B    DETAILS OF EXPERIMENTAL SETUP

All experiments are implemented in PyTorch and simulated in NVIDIA GeForce RTX 3090 GPUs. Codes for the results in this paper are provided in the supplementary material.

## B.1    DATASETS AND MODELS

We consider image classification tasks and evaluate our method on four popular datasets: (1) EMNIST (Extended MNIST) is a 62-class image classification dataset, extending the classic MNIST dataset (Cohen et al., 2017). It contains 62 categories of handwritten characters, including 10 digits, 26 uppercase letters and 26 lowercase letters; (2) Fashion-MNIST with 10 categories of clothes (Xiao et al., 2017); (3) CIFAR-10 with 10 categories of color images (Krizhevsky & Hinton, 2009); and (4) CINIC-10 (He et al., 2020), which is more diverse than CIFAR-10 as it is constructed from two different sources: ImageNet and CIFAR-10. We construct two different CNN models for EMNIST/Fashion-MNIST and CIFAR-10/CINIC-10, respectively. The first CNN model is constructed by two convolution layers with 16 and 32 channels respectively, each followed by a max pooling layer, and two fully-connected layers with 128 and 10 units before softmax output. And we use the LeakyReLU (Xu et al., 2015) activation function. The second CNN model is similar to the first one but has one more convolution layer with 64 channels.

## B.2    DATA PARTITIONING

Similar to (Karimireddy et al., 2020b; Zhang et al., 2021b; Huang et al., 2021), we make all clients have the same data size, in which $s\%$ of data (20% by default) are uniformly sampled from all classes and the remaining $(100 - s)\%$ from a set of dominant classes for each client. We evenly divide all clients into multiple groups while in each group the clients have the same dominant classes. Specifically, for Fashion-MNIST, CIFAR-10 and CINIC-10 datasets which have 10 categories of images, we divide the clients into 5 groups and use **three consecutive** classes as the dominant class set for each group, where the starting class is 0, 2, 4, 6 and 8 for the 5 groups, respectively. For EMNIST dataset, we divide the clients into 3 groups and use the **digits/uppercase/lowercase** as the dominant set for each group, respectively.

## B.3    IMPLEMENTATION DETAILS

**Training Settings.** We employ the min-batch SGD as the local optimizer for all approaches. The step size $\eta$ of local training is set to 0.01 for EMNIST/Fashion-MNIST, and 0.02 for CIFAR-10/CINIC-10. Notice that our method alternatively optimizes the feature extractor and the classifier. To reduce the local computational overhead, we only train the classifier for one epoch with a larger step size $\eta_g = 0.1$ for all experiments, and train the feature extractor for multiple epochs with the same step size $\eta_f = \eta$ as other baselines. The weight decay is set to 5e-4 and the momentum is set to 0.5. The batch size is fixed to $B = 50$ for all datasets except EMNIST, where we set $B = 100$. The number of local training epochs is set to $E = 5$ for all federated learning approaches unless explicitly specified. And the number of global communication rounds is set to 200 for all datasets, where all FL approaches have little or no accuracy gain with more communications. For all methods, we report the average test accuracy of all local models for performance evaluation.

**Compared Methods.** We compare our proposed FedPAC with the following approaches: Local-only, where each client trains model locally without collaboration between clients; model interpolation based method APFL (Deng et al., 2020); multi-task learning based methods, including pFedMe (Dinh et al., 2020) and Ditto (Li et al., 2021c); parameter decoupling based methods, including LG-FedAvg (Liang et al., 2020) with local feature extractor and global output layers, FedPer (Arivazhagan et al., 2019) and FedRep (Collins et al., 2021) that learn personal classifier on top of a shared feature extractor, FedBABU (Oh et al., 2022) that keeps the global classifier unchanged during the feature representation learning; ensemble methods that utilize both global and local models, including Fed-RoD (Chen & Chao, 2022) and kNN-Per (Marfoq et al., 2022); FedFomo (Zhang et al., 2021b) that performs weighted combination of local models, and pFedHN (Shamsian et al., 2021) that employs a server-side hypernetwork to generate personalized models.

**Hyper-parameters.** For FedPAC, we tune the penalty coefficient $\lambda$ over $\{0.1, 0.5, 1, 5, 10\}$ and fix it to 1 for all settings. For APFL, we tune the parameter $\alpha$ over $\{0.25, 0.5, 0.75\}$ and set 0.25 for

all settings; For pFedMe, we search $\lambda$ over $\{0.1, 1, 10, 100\}$ and set to 10, and also choose $K = 5$ for all settings; For, Ditto, we tune the parameter $\lambda$ among $\{0.1, 1, 10\}$, and use 1 for all settings. For kNN-Per, we tune the interpolation parameter over $\{0.5, 0.6, 0.7, 0.8\}$. For FedFomo, we set the number of models downloaded $M = 5$ as recommended in the paper. For pFedHN, we tune the learning rate over $\{0.1, 0.01, 0.001\}$ and then fix both the server and client learning rate to 0.01.

## C ADDITIONAL EXPERIMENTAL RESULTS

### C.1 PERSONALIZED CLASSIFIER COMBINATION WEIGHT

In this part, we investigate the classifier combination in FedPAC by estimating optimal client-to-client weights overtime, visualizing the calculated coefficient matrix during training process in Figure 2 and Figure 3 for EMNIST and CIFAR-10 datasets, respectively. We depict clients in the same group next to each other (e.g. clients 0, 1, 2, 3 belong to group 1 in CIFAR-10 setup). The weights are calculated according to A.2-A.3 and recorded in round 1, 20 and 50 as arranged from left to right in each subfigure. The results demonstrate that our method can reliably encourage clients with similar data distributions to collaborate more and prevent collaboration between unrelated clients. And it can be found that when data distributions across clients are highly heterogeneous, each client tends to collaborate only with those clients belonging to the same group, such as Figure 2(a)(b) and Figure 3(a)(b). When the data distributions are more homogeneous, each client will perform the combination of classifiers from all the other clients to obtain a better personalized one, as in Figure 2(d) and Figure 3(d).

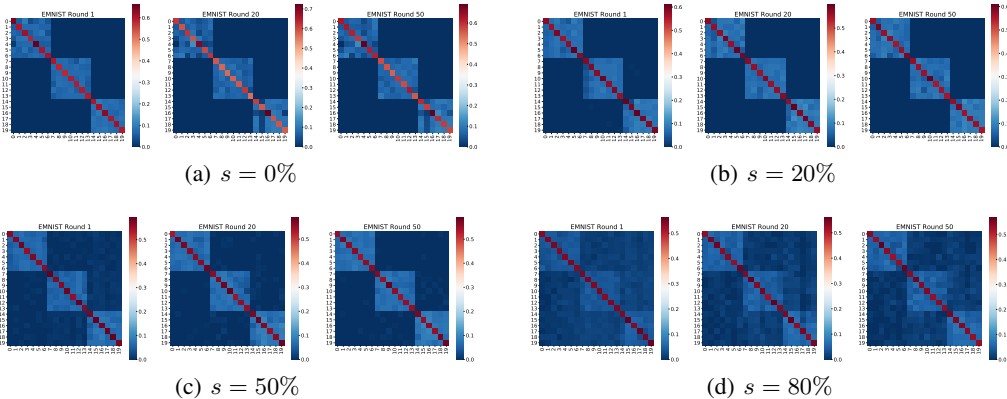

(a) $s = 0\%$          (b) $s = 20\%$

(c) $s = 50\%$          (d) $s = 80\%$

Figure 2: Classifier combination weights on EMNIST dataset under various levels of data heterogeneity.

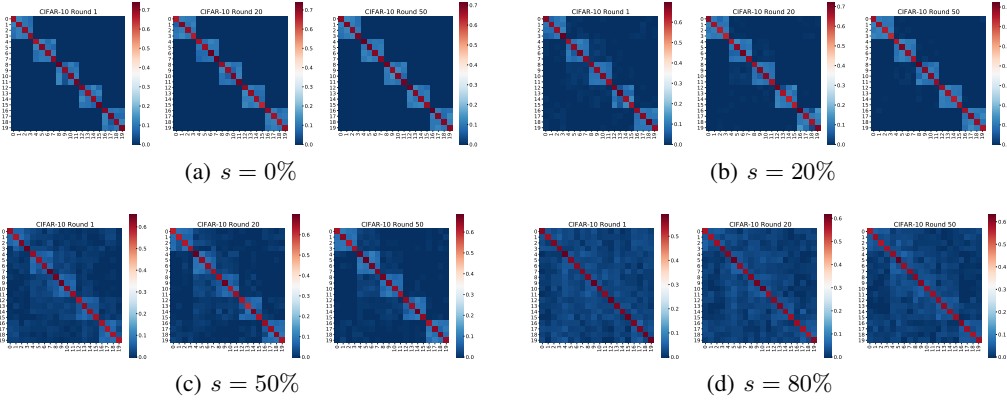

(a) $s = 0\%$          (b) $s = 20\%$

(c) $s = 50\%$          (d) $s = 80\%$

Figure 3: Classifier combination weights on CIFAR-10 dataset under various levels of data heterogeneity.

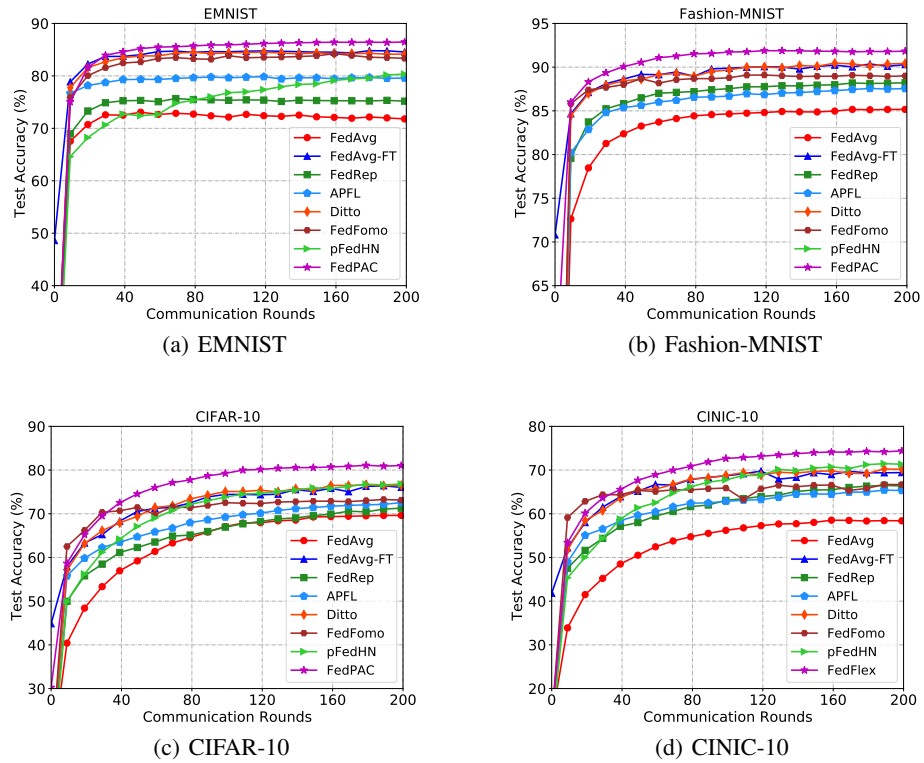

Figure 4: Test accuracy over communication rounds under different training methods for FL with 20 clients.

## C.2 CURVES OF TEST ACCURACY DURING TRAINING PROCESS

Figure 4 presents the evolution of average test accuracy over global communication rounds for partial experiments shown in Table 1. From which, it can be seen that our method almost outperforms other baselines during the whole training process, except the FedFomo has better model accuracy in the initial training phase. Besides, the accuracy curves of FedAvg with local fine-tuning almost overlap with Ditto, and result in competitive performance as other start-of-the-art methods, which demonstrate that FedAvg with local fine-tuning is simple yet effective, and tough-to-beat method.

## C.3 EFFECT OF LOCAL EPOCHS.

To reduce communication overhead, the clients tend to have more local training steps, which can lead to less global communication and thus faster convergence. Therefore, we monitor the behavior of FedPAC using a number of values of $E$, whose results are given in Figure 5. The results show that larger values of $E$ benefit the convergence speed and never hurt the model accuracy under our framework. Nevertheless, there is a trade-off between the computations and communications, i.e., while larger $E$ requires more computations at local clients, smaller $E$ needs more global communication rounds to converge.

## C.4 COMPARISON WITH SINGLE/MULTIPLE GLOBAL MODEL(S) BASED METHODS

In this part, we compare our method with improved FL algorithms that only learn a single global model and the multiple-models based approaches. We choose the two popular baselines in conventional FL, i.e., FedProx (Li et al., 2020b) and SCAFFOLD (Karimireddy et al., 2020b), to learn a single global model. For methods learning multiple global models, we evaluate the HypCluster in (Mansour et al., 2020) and the recently proposed FedEM (Marfoq et al., 2021). We construct two experimental settings, where the number of global models is set as 3 and 5, respectively. It is worth noting that we may not know the amount of latent distributions in advance, therefore it could be non-trivial to select the number of model components or clusters for striking a balance between communication/computation

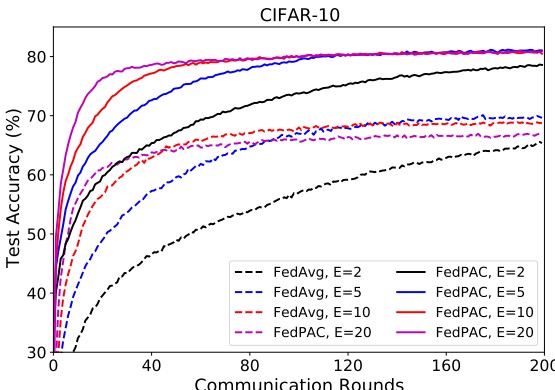

Figure 5: Performance comparison in average test accuracy on CIFAR-10 dataset with varying local epochs.

overhead and the performance of personalized models. We report the results in the following Table 4, where we also evaluate the local fine-tuned version of FedEM method (FedEM-FT), which is not applied in the original paper. From the results we can find that FedEM with local fine-tuning can lead to satisfactory performance in most cases at the expense of increased communication and computation costs, and the high performance could be attributed to the model ensemble, which however would also increase the computational burden during local inference.

Table 4: The comparison of final test accuracy (%) on different datasets. We apply full participation for FL system with 20 clients, and apply client sampling with rate 0.3 for FL system with 100 clients.

| Method | EMNIST | | Fashion-MNIST | | CIFAR-10 | | CINIC-10 | |
|---|---|---|---|---|---|---|---|---|
| | 20 clients | 100 clients | 20 clients | 100 clients | 20 clients | 100 clients | 20 clients | 100 clients |
| FedProx | 72.38 | 75.92 | 85.20 | 86.79 | 69.52 | 73.03 | 58.19 | 61.91 |
| SCAFFOLD | 72.66 | 76.68 | 86.29 | 89.31 | 72.56 | 77.22 | 62.58 | 70.98 |
| HypCluster ($k$=3) | 74.52 | 87.68 | 88.63 | 90.08 | 70.16 | 78.91 | 64.95 | 70.18 |
| HypCluster ($k$=5) | 83.30 | 88.04 | 89.08 | 91.50 | 73.80 | 79.57 | 67.13 | 71.38 |
| FedEM ($M$=3) | 73.48 | 75.77 | 85.97 | 87.43 | 72.44 | 77.01 | 62.82 | 65.34 |
| FedEM-FT ($M$=3) | 83.28 | 86.65 | 90.36 | 91.54 | 79.69 | 82.46 | 74.21 | 77.02 |
| FedEM ($M$=5) | 74.83 | 77.85 | 86.09 | 87.96 | 73.41 | 76.63 | 63.32 | 66.21 |
| FedEM-FT ($M$=5) | 83.42 | 86.72 | 90.56 | 91.59 | 80.17 | 83.15 | 74.47 | **78.35** |
| **FedPAC** (ours) | **86.46** | **89.88** | **91.83** | **92.72** | **81.13** | **83.36** | **74.96** | 77.36 |

## C.5    COMPARISON WITH FEDPROTO

We notice that a recent work FedProto (Tan et al., 2021b) also applied a similar regularization for model training. However, we find that only sharing global prototypes without the shared feature extractor would exhibit little performance improvement as shown in the following table. We evaluate the FedProto with locally trained classifiers in our experimental settings, where *FedProto-G* means using global prototype to regularize the local training without model exchanging and *FedProto-L* means only using local prototype without any communication. From the results we could find that both of them lead to similar results and cannot even outperform FedAvg-FT.

In this paper, the feature alignment in our method is inspired by our analyses, which focus on the principle of minimizing the testing loss and illuminates that the local-global feature distribution discrepancy would increase the generalization error. The theoretical result is consistent with the intuition so that we can understand why feature alignment would make sense. Consequently, we apply the similar local regularization term to force the model to learn more compact intra-class features. And it is also easy to understand why use local prototype to substitute global prototype can also

Table 5: Comparison of test accuracy (%) with FedProto. We set 20 clients with full participation.

| Dataset | Local-only | FedAvg-FT | FedProto-G | FedProto-L | **FedPAC** |
|---|---|---|---|---|---|
| Fashion-MNIST | 85.68 | 90.47 | 87.47 | 87.45 | **91.83** |
| CIFAR-10 | 65.43 | 76.56 | 67.48 | 67.45 | **81.13** |

achieve similar performance. From this point, the contribution in the part of feature alignment may not lie in the algorithm implementation but in the interpretation and understanding.

## C.6   RESULTS IN DATA IMBALANCED SETTING

For the main experiments, the training sample sizes are balanced for different clients and it is of significant interest to investigate the effect of imbalanced data sizes for more heterogeneous settings. Instead of the fixed size of 600 samples per client, here we randomly choose the size from $\{300, 600, 1200\}$ for each client regardless the local label distributions. It is worth noting that we focus on the relatively mild imbalanced setting in consideration of the potential straggler issues in FL when data sizes are highly imbalanced, which could be a problem of interest to investigate separately. We evaluate various methods on the Fashion-MNIST/CIFAR-10 and show the results in table 6, from which we can find that our method is robust to the quantity-skew and still outperforms other baselines.

Table 6: Comparison of test accuracy (%) under imbalanced setting with 20 clients and sampling rate 1.0.

| Dataset | Local-only | FedAvg-FT | FedRep | Ditto | Fed-RoD | pFedHN | HypCluster | FedEM | **FedPAC** |
|---|---|---|---|---|---|---|---|---|---|
| Fashion-MNIST | 85.65 | 90.75 | 90.13 | 90.71 | 89.95 | 87.67 | 88.36 | 90.72 | **92.33** |
| CIFAR-10 | 64.68 | 77.32 | 71.28 | 78.32 | 77.48 | 78.12 | 73.43 | 79.26 | **81.58** |

## C.7   RESULTS IN PATHOLOGICAL NON-IID SETTING

As a supplement, we also add some results on the pathological non-IID setting with 20 clients, where each group has different number of classes. More precisely, there are 3/3/5/5/8 different classes for clients in each specific group and training samples are balanced across classes at each client. We conduct two sets of experiments with different class allocations, where the local sample size is still fixed to 600 for each client. In the first set of experiments, the class set is randomly selected for each client and could be varied even for clients in the same group, while in the second set of experiments, the class sets differ across groups but are shared for clients belonging to the same group. We take the CIFAR-10 as an example and compare our method with several baselines, reporting the results in the following table. From the numerical results we could conclude that even though the hardness of local task varies across clients when the number of classes is different, our method still consistently achieves the best averaged accuracy attributed to the feature alignment and classifier collaboration mechanisms, which demonstrates the wide utility of our proposed framework in label distribution skew scenarios.

Table 7: Test accuracy (%) under pathological setting of CIFAR-10 with 20 clients and sampling rate 1.0.

| Class Allocation | Local-only | FedAvg-FT | FedRep | Ditto | Fed-RoD | pFedHN | HypCluster | FedEM | **FedPAC** |
|---|---|---|---|---|---|---|---|---|---|
| Random | 67.57 | 77.42 | 72.71 | 77.59 | 74.85 | 77.23 | 66.52 | 80.26 | **80.53** |
| Clustered | 64.62 | 75.85 | 69.76 | 76.25 | 73.58 | 77.51 | 72.98 | 78.71 | **79.65** |

## C.8   RESULTS IN DIRICHLET ALLOCATION SETTING

We also apply another widely used non-IID data partition strategy, i.e., distributing samples with the same label across the clients according to a Dirichlet distribution with a concentration parameter $\beta$. Specifically, we sample $p_k \sim Dir_m(\beta)$ and allocate a $p_{k,i}$ proportion of the instances of class

$k$ to client $i$. In this work, we set $\beta = 1.0$ and consider a FL setup with 100 clients, evaluating our proposed method as well as other baseline methods on the Fashion-MNIST and CIFAR-10 datasets. The test accuracy results are recorded in the following Table 8, from which we can see that our method again outperforms the other baselines, exhibiting the highest performance in both cases.

Table 8: Comparison of test accuracy (%) under Dirichlet allocation with 100 clients and sampling rate 0.3.

| Dataset | Local-only | FedAvg-FT | FedRep | Ditto | FedFomo | pFedHN | HypCluster | FedEM | **FedPAC** |
|---|---|---|---|---|---|---|---|---|---|
| Fashion-MNIST | 82.76 | 90.32 | 86.78 | 89.89 | 87.78 | 89.32 | 89.45 | 89.57 | **91.43** |
| CIFAR-10 | 34.68 | 61.88 | 44.54 | 73.53 | 56.39 | 75.74 | 73.69 | 76.79 | **78.53** |

## C.9 RESULTS IN HIGHLY DATA-SCARCE CASE

Notice that previous work usually partitions all the samples from the whole dataset to clients, while we only assign a fixed number of samples to each client. Thus, the amount of training samples for each client in our work is not such large, which is also the main motivation for classifier collaboration. Consequently, the Local-only method cannot achieve satisfactory performance. If local data size is sufficiently large, stand-alone training or FedAvg-FT should achieve similar accuracy with our method. Here we also provide additional experimental results for a relatively higher data-scarce case. We evaluate several competitive baselines on CIFAR-10 with 1000 clients, where we allocate only 50 samples (mini-batch size) for each client with default partition strategy and randomly select 30 clients in each round. The results listed in the following table clearly indicate that our method outperforms others with more significant improvement.

Table 9: Test accuracy (%) on CIFAR-10 with 1000 clients and sampling rate 0.03.

| Method | Local-only | FedAvg-FT | FedRep | Ditto | **FedPAC** |
|---|---|---|---|---|---|
| **Acc(%)** | 44.91 | 61.12 | 47.45 | 61.32 | **68.56** |

## C.10 RESULTS IN FEATURE SKEW SETTING

It is worth noting that our framework for classifier collaboration actually does not need the assumption of shared class-conditional distributions, which means it can automatically distinct those classifiers that can promote the performance for the target client and exclude the irrelevant ones. One example is the concept shift setting provided in the main text. Now, we provide another example of the setting with feature distribution skew, where we rotate the images in each group with a certain degree. We compare some methods on Fashion-MNIST and from the results we can find that our method still achieves the best performance.

Table 10: Test accuracy (%) on Fashion-MNIST under feature skew setting with 20 clients.

| Method | Local-only | FedAvg-FT | FedRep | Ditto | FedBABU | Fed-RoD | **FedPAC** |
|---|---|---|---|---|---|---|---|
| **Acc(%)** | 78.53 | 81.76 | 79.86 | 82.16 | 81.29 | 80.27 | **85.56** |

