# OpenReview forum: "Personalized Federated Learning with Feature Alignment and Classifier Collaboration"
_ICLR.cc/2023/Conference — ICLR 2023 notable top 5%_

### Official Review · Reviewer_ivCQ · 2022-10-24

**Confidence:** 3
**Correctness:** 4
**Technical Novelty And Significance:** 4
**Empirical Novelty And Significance:** 4
**Recommendation:** 8

**Clarity, Quality, Novelty And Reproducibility:**

The paper is mostly well-written and the presented results are satisfactory. The evaluation is carried out on public datasets, which can facilitate future comparisons with novel methods. An ablation study is also given to evaluate the contribution of the different losses proposed by the paper. In general, the proposed DREAM framework is not difficult to reproduce.

**Strength And Weaknesses:**

Strength

The structure of the proposed FedPAC is reasonable and easy to follow.
The studied problem (personalized federated learning) is interesting and different applications may benefit from it.
the experiment results are extensive and impressive
The authors also give a theoretical analysis.

Weaknesses

The description of motivation could be improved.
The paper could benefit from describing personalized federated learning in more detail.

**Summary Of The Paper:**

The authors propose FedPAC for personalized federated learning, which aims to learn a better representation by leveraging global semantic knowledge. Comprehensive experimental results multiple benchmarks demonstrate the effectiveness of the proposed framework.

**Summary Of The Review:**

I am generally optimistic about this submission. Considering the well-motivated problem, novel framework and impressive experimental results, I have no negative comments on the quality of the paper.

---

> ### Author Response · Authors · 2022-11-09
> **Response to Reviewer ivCQ**
>
> We sincerely thank the reviewer for the positive comments and helpful suggestions. We will try to strengthen the descriptions of the details and motivations of personalized federated learning in the revision soon.

---

### Official Review · Reviewer_Ddph · 2022-10-26

**Confidence:** 5
**Clarity, Quality, Novelty And Reproducibility:** The writing part should be improved, …
**Correctness:** 3
**Technical Novelty And Significance:** 2
**Empirical Novelty And Significance:** 2
**Recommendation:** 5

**Strength And Weaknesses:**

Strength:

1. The improvement of experimental results is significant.

Weaknesses:

1. The description of some details in the proposed method is not clear.

2. Need to make comparisons with some recent PFL works to verify the effectiveness.


**Summary Of The Paper:**

This work solves personalized federated learning (PFL) setting by developing feature alignment mechanism and classifier collaboration. Concretely, this feature alignment minimizes the distance between local feature representations and global centroids. And classifier collaboration utilizes linear combination of multiple local classifiers to customize new classifier. The current work conducts experiments on four datasets to verify the effectiveness of proposed method.

**Summary Of The Review:**

This work solves personalized federated learning (PFL) setting by developing feature alignment mechanism and classifier collaboration. Concretely, this feature alignment minimizes the distance between local feature representations and global centroids. And classifier collaboration utilizes linear combination of multiple local classifiers to customize new classifier. The current work conducts experiments on four datasets to verify the effectiveness of proposed method.

However, I have several concerns on this work.

1. Eq. (1) is the optimization goal of traditional federated learning (FL) instead of that of PFL task.

2. The feature alignment regularization needs to access the global centroids which are obtained by aggregating local centroids. Under this condition, the data distribution information of each client has been leaked out, which violates the requirement of privacy protection in PFL task.

3. In classifier collaboration, why the simple linear combination of local classifiers can find these clients with the similar distributions? It would be better to analyze the combination coefficients to support this point.

---

> ### Author Response · Authors · 2022-11-10
> **Response to Reviewer Ddph**
>
> We thank the reviewer for the time and effort spent in reviewing as well as the useful comments and suggestions. The following is our response.
>
> **Q1:** Eq. (1) is the optimization goal of traditional federated learning (FL) instead of that of PFL task.
>
> **A1:** We are sorry for making the reviewer and possible readers confused, actually the notation $\mathbf{w}$ used in Eq. (1) denotes the collection of local model parameters, $\mathbf{w}=(\mathcal{w}_1,{\mathcal{w}}_2,...,\mathcal{w}_m)$,  instead of a single global model, maybe it is better to use the capital $\mathbf{W}$ as the notation of model set. We will submit a revision with more clear description.
>
>
> **Q2:** The feature alignment regularization needs to access the global centroids which are obtained by aggregating local centroids. Under this condition, the data distribution information of each client has been leaked out, which violates the requirement of privacy protection in PFL task.
>
> **A2:** It is possible that a client may or may not be willing to share the data distribution information to the server. In such cases, a client could choose to apply the local centroids to substitute the global ones and give up both the classifier combination and the regularization w.s.t the global centroids. As a result, the performance of personalized model might be reduced. However, if more lightweight information could be shared, a client could more easily to find the “friend” clients to help improve the classifier learning and thus benefit the local performance.
>
> We also notice that transmission of local feature centroids/prototypes with distribution information is considered reasonable in previous image classification tasks as in [1-2], where different clients have different label distributions. Depending on the specific tasks and applications, class-wise feature statistics and data distribution information may or may not be considered sensitive information. In this work, our focus is to effectively address the challenges of learning better feature extractor and conducting classifier collaboration to boost personalized model utility in federated framework. To this end, we utilize the global feature centroid with minimal additional privacy risk. We also want to point out that our proposed framework can be integrated with various privacy-preserving techniques such as secure aggregation and trusted execution environment to further enhance the reliability of the system.
>
> Finally, we would like to point out that the trade-offs between utility, security and efficiency still exist in personalized FL. Our framework mainly focuses on improving the model utility, which is also the main incentive for a client to participate the personalized federated learning instead of stand-alone training, and even to share more information for a better collaboration.
>
>
> [1] FedProto: Federated Prototype Learning across Heterogeneous Clients, AAAI 2022
>
> [2] Federated Learning from Pre-Trained Models: A Contrastive Learning Approach, NeurIPS 2022
>
> &nbsp;
>
> **Q3:** In classifier collaboration, why the simple linear combination of local classifiers can find these clients with the similar distributions? It would be better to analyze the combination coefficients to support this point.
>
> **A3:** Even for the simple linear combination, it is not easy to analyze and compute the optimal combining coefficients. One of the key contributions of our paper is using the theoretical analysis to provide a possible way to characterize the optimal coefficients in an analytical way and estimating the coefficients by solving a quadratic optimization problem, instead of heuristic methods. The details are provided in the Appendix A2-A5.
>
> The formulated quadratic optimization problem in this paper is related to both the local sample sizes and label distributions, and exhibits a bias-variance trade-off during the collaboration. The locally trained classifier is unbiased but could have high variance due to limited local samples, while the classifiers from other clients might be either biased or unbiased with varied variances. Therefore, incorporating the “friend” clients with proper coefficients is indeed striking a balance in the bias-variance trade-off. And in this paper, we provide a computable and feasible way to solving this trade-off. Some visualization results in Appendix C.1 clearly show that our method could reliably find the friendly clients and prevent negative collaboration between unrelated clients.

---

> > ### Comment · Reviewer_Ddph · 2022-11-30
> > **Thanks for authors' explanations.**
> >
> > After reading authors' rebuttal, most of my concerns are solved. But the data privacy leakage is still the inevitable drawback when using the proposed method in this paper. Although this rebuttal claims that the proposed method can be integrated into some privacy-preserving techniques, it is not clear that how to implement the integration. Could you provide the specific references of the mentioned privacy-preserving techniques? Thus, it would be better to discuss this point in the final version if this work will be accepted.

---

> > > ### Author Response · Authors · 2022-12-01
> > > **Response to Comment from Reviewer Ddph**
> > >
> > > We thank the reviewer for the valuable comment and we provide some discussions in the following.
> > >
> > >
> > > We think that there is indeed a tension between the data privacy and model performance. In the context of personalized FL, the model performance is the main incentive for a client to participate the FL instead of stand-alone training, and sharing more non-sensitive information can lead to a better collaboration. In FL, the primary goal is prohibiting the exposure of raw data features. If the data distribution information is also regarded as sensitive information, then, without privacy protection circumstances, it is hard to strictly guarantee the privacy even for the vanilla FedAvg that only aggregates local models without sharing raw data. For example, previous works [1,2] empirically show some evidences that the classifier layers may contain some information of local class distributions. Thus, federated learning would need more privacy-preserving techniques to enhance the system reliability in practice, e.g., secure aggregation [3] and homomorphic encryption (HE) [4].
> > >
> > > Take the homomorphic encryption as example, the clients receive private keys to homomorphically encrypt their model parameters before sending them to the server. The server doesn’t own a key and only sees the encrypted models. But the server can aggregate these encrypted model parameters and then send the updated model back to the clients. The clients can decrypt the model parameters because they have the keys and can then continue with the next round of local training.
> > >
> > > For our framework, where not only local models but also category-level feature statistics are transmitted, the aggregation of models/centroids can be protected by applying HE techniques while the true category-level feature statistics could be directly used to compute the personalized combination coefficients. We acknowledge that to further quantify the potential leakage from feature centroids would involve in designing specific inversion attacks, which is out of the scope of our work. But we consider that the additional privacy risk may not be an issue when only category-level feature statistics are provided to server while the model parameters are encrypted. Without accessing the feature extractor layers, the server is hard to infer meaningful semantic data information from only the feature centroids and the trace of feature covariance.
> > >
> > > The above discussions are all based on the semantically aligned label across clients, as discussed in the paper, our framework can also be extended/adapted into the case of concept shift, where the label is meaningless for the server as each label doesn’t have a certain meaning. In such cases, we use the local centroids to substitute the global centroids while the classifier collaboration can be conducted as usual.
> > >
> > > [1] No Fear of Heterogeneity: Classifier Calibration for Federated Learning with Non-IID data, NeurIPS 2021
> > >
> > > [2] FedRS: Federated Learning with Restricted Softmax for Label Distribution Non-IID Data, KDD 2021
> > >
> > > [3] Practical Secure Aggregation for Privacy-Preserving Machine Learning, CCS 2017
> > >
> > > [4] BatchCrypt: Efficient Homomorphic Encryption for Cross-Silo Federated Learning, USENIX ATC 2020

---

### Official Review · Reviewer_8wN3 · 2022-11-02

**Confidence:** 4
**Correctness:** 3
**Technical Novelty And Significance:** 3
**Empirical Novelty And Significance:** 3
**Recommendation:** 8

**Clarity, Quality, Novelty And Reproducibility:**

The paper proposes a novel idea and the presentation is good. There may exist some flaws mentioned above.

**Strength And Weaknesses:**

**Strength**
1. The proposed framework is novel. The decomposition of feature learning and linear classification helps customize local clients.
2. The performance is theoretically analyzed based on a simplified model and numerically validated by experiments on several datasets.

**Weakness**
1. The heterogeneity of data considered in this paper is a specialized one. Generally, the heterogeneous local data distributions will be different in both total number of samples and number of feasible label classes. Current setting assumes each client has the same size of data and observes the full set of label classes. It is interesting to see the performance of the proposed method when the local data distributions are heterogeneous enough.
2. In Eq. (3), the authors assume there only exists one centroid for each label class. This assumption needs to be better explained. It is likely that samples in one label class may come from two different centroids. Simply averaging all the features as one centroid may not always be true.
3. In Eq. (4), the authors mention $\alpha$ is optimized to reduce the local testing loss. It is not clear whether the local test data is different from the test data used to show Table 1.





**Summary Of The Paper:**

The paper proposes to deal with the data heterogeneity in federated learning by local-global feature alignment and weighted combination of local classifiers. The whole model is decomposed into two parts, a feature extractor $f$ and a linear classifier $g$ built on $f$. The feature alignment is achieved by extracting feature centroid for each label class. The parameters of each local classifier $g_i$ is combined with other classifiers by optimized weights so as to bundle similar local clients.

**Summary Of The Review:**

The paper is overall well-written but there are some critical issues that need to be addressed. I'm happy to increase my evaluation if my concerns (see weakness) are addressed.

----------- Post rebuttal ----------

All of my concerns are well-addressed. I increased my evaluation from 5 to 8.

---

> ### Author Response · Authors · 2022-11-10
> **Response to Reviewer 8wN3 (2/2)**
>
> **(A2 Continued)**
> - Different from FedRep, our framework not only conducts classifier-level collaboration but also explicitly aligns the feature distribution across clients by adding a regularization term w.s.t the global feature centroids, both of which are theoretical analysis inspired with interpretation. And our framework can be adapted to concept shift scenario by substituting the unknown global centroids with local centroids and it can automatically exclude those clients with concept shift during the classifier collaboration.
>
>
> - As one more possible extension of our framework, consider such a case where two groups of clients have the same number of labels, but the semantic meaning or feature distribution are varied, e.g., the first group learns the first 5 classes of CIFAR-10 and the second group learns the remaining 5 classes, both with a 5-class softmax classifier. Then, the personalized FL methods based on the adaptation of global model will be invalid and the clustered FL methods will tend to learn different global models for each group and thus could not benefit from the similar data structures of the other group. In contrast, our framework could automatically uncover the group structures and conduct intra-group linear classifier combination (global feature centroids could also be obtained in a group-wise manner or just substitute with the local centroids), while the shared feature extractor could benefit both learning tasks.
>
>
>
> **Q3:** In Eq. (4), the authors mention \alpha is optimized to reduce the local testing loss. It is not clear whether the local test data is different from the test data used to show Table 1.
>
> **A3:** Thanks for pointing it out and we agree that the description could be improved to make it better understood by readers. Actually, the local test data used in Table 1 is only a small empirical test set with the same distribution as the local training set, while the testing loss mentioned in the theoretical part indicates the expectation of empirical testing loss and could also be interpreted as the true testing loss over the underlying input space, which is actually not accessible. As the final goal of model learning is to minimize the testing loss instead of the training loss, otherwise the locally learned classifier would fit the training data best and no more collaboration is needed. And the obtained quadratic optimization problem in Appendix A.2 is related to some statistics of the true testing set. However, as mentioned before, the true testing loss is infeasible as we could not collect all possible samples, we turn to use the statistics of the training set to solve the optimization problem so as to obtain a reliable estimate of combining coefficients. For empirical performance evaluation, a small set of test set would be sufficient as previous works.

---

> > ### Comment · Reviewer_8wN3 · 2022-11-22
> > **Thanks for the additional experiments and explanations. Quick questions**
> >
> > The rebuttal resolved most of my concerns. But there is still one minor question wrt A3. If I understand correctly, some data in the test set is used to update $\alpha$. There are some questions related to this point.
> >
> > 1. Is it also implemented in existing works? Any reference?
> >
> > 2. Could you give more numerical details? For example, in CIFAR-10 with 100 clients, we will have 500 training examples for each client. How many test examples are used for doing a real test and how many test examples are used for updating $\alpha$ in this case?
> >
> > 3. Did you use the same test examples (you've used for updating $\alpha$) in minimizing the empirical loss for other methods compared in Table 1?

---

> > > ### Author Response · Authors · 2022-11-23
> > > **Response to Questions from Reviewer 8wN3**
> > >
> > > We thank the reviewer for this addtional valuable comment and question, and we hope the clarifications provided in the following could help address the questions/concerns.
> > >
> > > - In our evaluation, we allocate each client with a training set and a test set, which share the same distribution. However, the test samples are held out from the training procedure, and are only used to evaluate the performance of personalized models, only the training samples are involved throughout the training process.
> > >
> > > - To the best of our knowledge, some relevant works conducting personalized aggregation all assume that each client can have access to models from all the other clients and utilize an extra local validation set to learn the proper coefficients [1,2,3]. In contrast, our framework does not assume the local models sharing among clients, which may raise more severe communication and privacy issues, and our method proposed to characterize and solve the coefficient estimation in an analytical way instead of empirical searching via the validation/test data.
> > >
> > > - One may ask that how could the $\alpha$ be updated without the validation/test samples. If we rely on the learning-based methods as existing works, essentially, we need the validation samples to search out the helpful peer clients. However, our approach is based on the analytical solution, which only needs some low-dimensional feature statistics and those statistics could also be estimated from the training set, as the training set itself is already an unbiased sampling of the underlying true distribution.
> > >
> > > We summarize some clarifications about our framework as follows.
> > > 1. The quantities (some statistics of feature embeddings) in our formulated quadratic optimization problem are related to the true underlying distribution.
> > > 2. Thus, the original optimization problem is infeasible as the whole set of test data is inaccessible.
> > > 3. If we have a relatively large set of samples, according to the large number theorem, we could use this unbiased set to get a good estimate of those quantities and then solve the optimization problem.
> > > 4. In our implementation, we even do not rely on any extra validation/test sets and directly use the training set to estimate the necessary quantities (see Appendix A.3-A.5), this operation can relax the requirement of validation data and works well enough.
> > > 5. If extra validation set (not used for model test) exists (depending on the specific setting), of course it could be used to update the $\alpha$ as well.
> > >
> > > We consider that this point could help to show the advantages of our framework in estimating such combination coefficients for model aggregation. Again, many thanks to the reviewer for putting out this question.
> > >
> > >
> > > [1] Personalized Federated Learning with First Order Model Optimization, ICLR 2021
> > >
> > > [2] Adapt to Adaptation: Learning Personalization for Cross-Silo Federated Learning, IJCAI 2022
> > >
> > > [3] Learning to Collaborate in Decentralized Learning of Personalized Models, CVPR 2022

---

> > > > ### Comment · Reviewer_8wN3 · 2022-12-07
> > > > **Thanks for your clarifications**
> > > >
> > > > Thanks for the detailed clarifications. It would be better if an extra explanation is added for Eq. (9), e.g., "we calculate the empirical training loss in practice since the training samples in each client are assumed to be iid as the testing samples."
> > > >
> > > > I increased my evaluation to 8.

---

> > > > > ### Author Response · Authors · 2022-12-07
> > > > > **Thanks :)**
> > > > >
> > > > > The authors appreciate the valuable discussions with the reviewer, where the constructive comments and suggestions are indeed helpful for improving the paper! Also, many thanks for raising the score.

---

> ### Author Response · Authors · 2022-11-10
> **Response to Reviewer 8wN3 (1/2)**
>
> The authors sincerely thank the reviewer for the insightful and valuable comments. The following is our response.
>
> **Q1:**  How is the performance of the proposed method when the local data distributions are heterogeneous enough, e.g., different in both total number of samples and number of feasible label classes.
>
> **A1:** Thanks for the comment, this is an interesting direction to investigate the collaboration between clients with different numbers of feasible label classes. We have provided some discussions in the common response and will also make more clarifications here.
>
> For the theoretical analysis part, we simultaneously consider the local label distribution and sample size, and the resulted quadratic optimization problem is also related to both of them. For the experimental evaluation, we admit that the main results are based on clients with full label set and same sample sizes. We mainly consider that in the FL system if the quantities of samples differ largely across clients, there may exist the straggler issues, and we find that the impact of sample sizes during the coefficient computing is limited compared to the label distributions under the mild data imbalance. Therefore, we focus on the label distribution skew instead of the data quantities. For a more comprehensive evaluation, we will add more experiments with imbalanced local sample sizes in the revision. Nevertheless, to verify the general effectiveness, we also evaluate the case with Dirichlet distribution-based partition in Appendix C.6, where both the label distributions and sample sizes vary across clients. And the results clearly show that our method still outperform baseline methods.
>
>
> **Q2:** The assumption of one feature centroid for each label class. It is likely that samples in one label class may come from two different centroids.
>
> **A2:** We thank the reviewer for this very interesting and insightful comment, which we think could help understand the effectiveness of our proposed method and other general federated learning algorithms. Generally, to achieve the goal of classification by a linear classifier (e.g., softmax classifier), the model training will try to pull the samples belonging to the same label together and push away samples from other classes in the feature embedding space. Conceptually, more compact intra-class feature distribution and larger inter-class distribution distance will lead to better classification accuracy. And this also the desired training result of the commonly used softmax classifier and the cross-entropy loss.
>
> - For the raw training data, it is likely that samples in one label class may come from different centroids, even the extracted feature embeddings could come from different centroids, e.g., following a mixture of multiple Gaussian distributions. However, from the perspective of linear classification, even the class-wise features follow the Gaussian mixture model, they could be regarded as a big cluster and the decision boundary of the classifier should try to separate the feature clusters of different classes. And a better feature extraction should result in feature clusters with higher linear separability.
>
> - Now revisit the FedPer/FedRep methods, as the training of feature extractor is conducted locally and independently, as a result, only the limited local training samples are well separated across classes. It is also likely that samples in the same label may result in different feature centroids for different clients. And simply averaging the parameters of local feature extractors could not remedy this very well. This could partially explain the reason that the FedPer/FedRep would result in relatively low performance compared to the simple-yet-effective FedAvg with fine-tuning. As FedPer and FedRep entirely do not share the classifier layers and thus the local feature centroids on training samples of the same class could be diverse, even under the exactly same feature extractor, which also means the learned semantic features are not well aligned across clients.
>
> - It’s an interesting question to analyze why the vanilla FedAvg is more advantageous than the FedRep in representation learning. We give one possible explanation as follows. Note that in the cross-entropy loss of softmax model, the logits are computed as the inner product between the instance feature-embedding and the class-embedding (one dimension of the classifier), in FedAvg the classifier is also shared among clients, therefore the local representation learning process is also implicitly aligned by the shared class-embeddings. This also indicates one drawback of FedAvg in tackling the concept shift issue as the conflict labels make the classifier layer averaging meaningless and could deteriorate the local training.

---

### Official Review · Reviewer_zWd3 · 2022-11-04

**Confidence:** 3
**Correctness:** 4
**Technical Novelty And Significance:** 3
**Empirical Novelty And Significance:** 3
**Recommendation:** 8

**Clarity, Quality, Novelty And Reproducibility:**

The paper is clearly written and easy to follow. The work is original to my knowledge.

**Strength And Weaknesses:**

Strengths: The proposed method is well-motivated. The use of global feature centroids in local training is straightforward and practical, and the idea of forming classifiers combination coefficient selection as a quadratic programming problem is elegant. Evaluation and ablation studies seem convincing.

Weaknesses: The proposed framework tackles the specific label distribution skew challenge in federated learning. It would be nice if the authors could discuss whether the method generalizes to other challenges caused by heterogeneous data distributions in federated learning, for example, when the number of classes is different (if so, whether the method needs any adaptations; if not, why the method has certain limited applications).

**Summary Of The Paper:**

This paper proposes a new learning framework to tackle the label distribution skew challenge in federated learning. Specifically, the paper introduces a novel feature representation alignment technique and a classifier combination technique to exploit both shared representation and inter-client classifier collaboration. By using a local regularization term related to the global feature centroids of each class, the framework mitigates the drift of local features during the learning process. By introducing a linear combination of classifiers with optimized coefficients during training, the framework allows similar clients to share more information while avoiding negative transfer from more distinct clients.

**Summary Of The Review:**

This paper introduces global feature centroids in feature regularization and classifier combination weights selection through quadratic programming to improve personalized federated learning. The proposed method is backed by good intuitive interpretation, theoretical analyses, and convincing evaluation results. The ideas introduced in this paper could be inspiring to the relevant community.

---

> ### Author Response · Authors · 2022-11-09
> **Response to Reviewer zWd3**
>
> The authors sincerely thank the reviewer for the positive feedbacks and helpful suggestions. Please refer to the common response where  some discussions on the heterogeneous data distributions are provided. Many thanks!

---

### Author Response · Authors · 2022-11-09
**Common Response to Reviewer zWd3 and Reviewer 8wN3 (2/2)**

**(3)** Recall that the data heterogeneity could be caused by label distribution skew, feature distribution skew, imbalanced data size and concept shift across clients. To the best of authors’ knowledge, none of the existing PFL methods could handle all of above scenarios in one framework. Among which the label distribution skew is the most frequently studied case and is also our focus. An experimental study [1] shows that the label distribution skew case is the most challenging setting in FL compared to the feature distribution and quantity skew settings, which directly motivates the personalized models in FL. Existing methods either keep the final classification layer local [2-3] or perform local adaptation [4-5]. However, they cannot perform well when local training data is scarce due to local over-fitting. There are also some studies investigating the inter-client collaboration instead of training a global model [6-7].

The non-IID data partition used in this paper mainly follows the prior works [6-7], where both the pathological non-IID data setting (each client is randomly assigned a set of class) and a practical non-IID data setting (multiple latent distributions exist and unknown to the server) are considered. As we aim to conduct classifier collaboration between similar clients, we focus on the clustered structure of clients as the practical non-IID data setting. Moreover, the existing methods did not consider the concept shift case, while our framework can be easily extended to such scenarios and achieve better performance.

[1] Federated Learning on Non-IID Data Silos: An Experimental Study, ICDE 2022

[2] Federated learning with personalization layers. arxiv:1912.00818, 2019

[3] Exploiting Shared Representations for Personalized Federated Learning, ICML 2021

[4] Personalized Federated Learning with Moreau Envelopes, NeurIPS 2020

[5] On Bridging Generic and Personalized Federated Learning for Image Classification, ICLR 2022

[6] Personalized Cross-Silo Federated Learning on Non-IID Data, AAAI 2021

[7] Personalized Federated Learning with First Order Model Optimization, ICLR 2021

&nbsp;

**(4)** Finally, we provide some more discussions as follows.
- In our framework the coefficients are used to linearly combining the entire local classifiers, and are estimated by solving the proposed quadratic optimization problem. Therefore, the classifier-level collaboration will tend to include those clients with similar label distributions and exclude those have largely different label distributions, e.g., different number of labels or non-overlapping label sets.

- If the set of feasible label classes differs across clients but the feasible classes are balanced, we can divide this situation into some sub-cases for discussion. (1) if two clients have totally different and non-overlapping sets of labels, then those two clients should not perform classifier collaboration; (2) if two clients have exactly the same set of labels, then those two clients are encouraged to collaboration, which will be automatically achieved by our framework; (3) if two clients have partially overlapped sets of labels, directly combining the entire classifier layers may not be a good strategy, and our current framework will also result in small coefficients for this case.

- For the last case mentioned above, we think that more fine-grained collaboration strategy might be needed, e.g., class-wise combination for the classifier. However, this kind of collaboration may only remain in heuristics and is hard to conduct rigorous theoretical analysis and algorithm optimization as different dimensions/components of the trained classifier are coupled, therefore we leave it as the future work. Moreover, as mentioned before, the collaboration between similar clients already brings significant improvement and the collaboration between partially overlapped clients might be desired only when no other similar client exists.

- We also want to point out that we focus on the server-based system and assume that each client will only train a local classifier fitting its own local distribution. If one client has prior knowledge on the data distributions of other clients, it might be possible to utilize the relevant local data (samples belonging to the classes of target clients) to train a particular classifier for other individual clients, which is beyond our scope and itself could be an interesting direction.

---

### Author Response · Authors · 2022-11-11
**Common Response to Reviewer zWd3 and Reviewer 8wN3 (1/2)**

We sincerely thank the reviewers for the insightful comments and valuable suggestions. We provide some discussions on the heterogeneous data distributions in the following.

**(1)** Commonly, the labels in clients are unified by the central server and samples with same label belong to the same semantic object/class. The final classifier (a matrix with d*K, d is the dimension of feature and K is the total number of classes) is also unified by the server. In such cases, there is no concept shift and current FL algorithms are generally applicable. Under this condition, if each client could only observe part of classes and the number of classes possibly differs across clients, it is interesting to discuss whether the current framework still works.

We give a positive answer to this problem. Notice that our framework consists of two key components, including explicit feature alignment and classifier collaboration. The ablation studies in paper show that both of them have significant benefits. When no concept shift occurs, the feature alignment could always be feasible and help to learn better feature extractor, regardless the diversity of observed classes. And our quadratic optimization -based coefficient estimation method could automatically bundle the clients with the same set of classes and exclude the irrelevant clients.

- For example, the Figure 2-3 in the Appendix visualize the combinating coefficients under different data heterogeneity, where s=0% means each client only has samples from dominant classes and no samples of other classes, and our framework can automatically choose the clients with the same class set to conduct classifier collaboration.

- For the extreme case where each client will have its own unique set of classes, the worst result would only be no classifier collaboration between clients but each client still benefits from the feature alignment, which will not be worse than the FedPer/FedRep methods.

- One may also think that if the class sets of two clients are different but partially overlapped, ignoring the classifier collaboration would be wasteful. For this case, we consider that the global feature alignment and classifier combination between clients with same tasks can already bring significant improvement, while additional collaboration between clients with partially overlapped classes would only provide marginal benefit. Moreover, we think that dimension-wise fusion between two classifiers is necessary to achieve more flexible collaboration in such cases, which is however non-trivial and needs more theoretical and empirical investigations. We will leave it as the further work.

**(2)** If we consider a more general and challenging scenario where the labels in local clients are not unified by the server and the number of classes in local tasks may also differ from each other, then concept shift could exist and the local classifier layers could differ in dimensions. There is no reason that we should think the samples with the same label must belong to the same semantic object, and obviously the averaging-based FL algorithms are not applicable any more due to inconsistent local objectives and unmatched classifier dimensions.

- If local concept shift exists, even if the dimension of classifier is unified by the server, the traditional FL algorithms still suffer. We evaluate this case where different groups of clients have concept shift but the classifier dimensions are consistent. Since the concept shift exists, the global feature alignment is also not directly applicable as the server could not figure out the corresponding relationship between local labels. To ease this issue, we use the local feature centroid to substitute the unknown global centroids, and the results in Table 3 verified that our method can reliably distinct the inter-group peer clients and consistently outperform the baselines.

- If the dimension of classifier differs across clients, the current form of our framework will still only conduct collaboration between clients with the same classifier dimensions and automatically adjust the coefficients by the task similarity. To handle this challenge more flexibly, necessary adaptation and extension of our framework is required (so as to other methods). Consider a simple case with only two clients (one as source client and one as target client), the classifier combination strategy between them is not just computing a scalar coefficient anymore, we should turn to estimate a new coefficient matrix, where each row is a coefficient vector responsible for selecting the appropriate vector components from the source classifier. Depending on the prior knowledge the server has, the design and computing of the coefficient matrix could be varied. As mentioned before, we emphasize that this most general case is essentially different from current problem setting and beyond our scope in this paper. Therefore, we leave it as the further work of this paper.

---

### Author Response · Authors · 2022-11-15
**Paper Revison Submitted**

We sincerely thank the reviewers for their constructive comments and helpful suggestions.  We provide a paper revision in direct response to the reviewers’ valuable comments. Here are the main changes:

- Adding a new recently published method kNN-Per [1] for comparsion in Table 1

-  We add a new experiment (Table 6 in Appendix C.6) that evaluates the data imbalanced setting

-  We add addtional experiments (Table 7 in Appendix C.7) for the pathological non-IID setting with different numbers of classes across clients

- A relatively high data-scarce case with only 50 samples per client is evaluated (Table 9 in Appendix C.9)

- A simple feature-skew case with rotated images per group is evaluated to demonstrate the extendibility of our framework (Table 10 in Appendix C.10)


[1]  Personalized federated learning through local memorization, ICML 2022

---

### Public Comment · ~Seongyoon_Kim1 · 2023-04-19
**Public Github**

Hello, I enjoyed reading your paper and congratulations on its acceptance. Could you please share the GitHub code for the paper? Thank you.

---

> ### Author Response · Authors · 2023-04-21
> **Github code**
>
> Thanks for your interest, maybe you can check it out at https://github.com/JianXu95/FedPAC.

---

### Decision · Program_Chairs · 2023-01-20

**Decision:**

Accept: notable-top-5%

**Justification For Why Not Higher Score:**

A reviewer remained concerned about the data privacy leakage.

**Justification For Why Not Lower Score:**

Three reviewers rated it 8. They liked the paper's intuition, theoretical analyses, and experiments.

**Metareview: Summary, Strengths And Weaknesses:**

Four experts reviewed the paper and provided ratings of 8, 8, 8, and 5, respectively. The reviewer who remained negative post-rebuttal was mainly concerned about data privacy leakage, but the other reviewers were satisfied with the rebuttal. Therefore, AC decided to side with the majority of reviewers as there was no further response from the negative reviewer. Hence, the decision is to recommend the paper for acceptance. The authors are encouraged to make the necessary changes to the paper to the best of their ability following the reviewers' comments.

**Note From Pc:**

if the above contains the word "oral" or "spotlight" please see: "oral" presentation means -> notable-top-5% and "spotlight" means -> notable-top-25%. As stated in our emails, we are disassociating presentation type from AC recommendations